# Defining morphologically and genetically distinct GABAergic/cholinergic amacrine cell subtypes in the vertebrate retina

Yan Li[1,2], Shuguang Yu[1,2], Xinling Jia[1], Xiaoying Qiu[1], Jie He [1] *

**1** Institute of Neuroscience, State Key Laboratory of Neuroscience, Center for Excellence in Brain Science and Intelligence Technology, Chinese Academy of Sciences, Shanghai, China, **2** University of Chinese Academy of Sciences, Beijing, China

* jiehe@ion.ac.cn

**Data Availability Statement:** All relevant data are within the paper and its Supporting Information files.

**Funding:** This study was funded by grants from the National Key Research and Development Program

## Abstract

In mammals, retinal direction selectivity originates from GABAergic/cholinergic amacrine cells (ACs) specifically expressing the *sox2* gene. However, the cellular diversity of GABAergic/cholinergic ACs of other vertebrate species remains largely unexplored. Here, we identified 2 morphologically and genetically distinct GABAergic/cholinergic AC types in zebrafish, a previously undescribed *bhlhe22*+ type and a mammalian counterpart *sox2*+ type. Notably, while sole *sox2* disruption removed *sox2*+ type, the codisruption of *bhlhe22* and *bhlhe23* was required to remove *bhlhe22*+ type. Also, both types significantly differed in dendritic arbors, lamination, and soma position. Furthermore, *in vivo* two-photon calcium imaging and the behavior assay suggested the direction selectivity of both AC types. Nevertheless, the 2 types showed preferential responses to moving bars of different sizes. Thus, our findings provide new cellular diversity and functional characteristics of GABAergic/cholinergic ACs in the vertebrate retina.

## Introduction

In the vertebrate retina, direction-selective retinal ganglion cells (dsRGCs) can respond to moving gratings at preferential directions (PDs) [1,2]. This direction selectivity originates from asymmetric dendritic inhibition of dsRGCs from GABAergic starburst amacrine cells (SACs), which are also cholinergic [3–8]. Isotropic cholinergic excitation and anisotropic GABAergic inhibition from SACs are essential for the direction selectivity of dsRGCs for moving gratings with low contrasts [9]. Thus, coexpression of GABAergic and cholinergic markers represents the neurotransmitter characteristic of SACs. In the mouse retina, SACs comprise ON and OFF subtypes, whose cell bodies are displaced in the RGC layer (GCL) and placed in the inner nuclear layer (INL), respectively [3,10,11]. In the mouse retina, the expression of *Sox2* and *Fezf1* was required for appropriate cell body positioning of ON subtype SACs [12,13]. The early study of the mouse retina showed that OFF, but not ON, subtype SACs received direct GABAergic inputs from non-SACs. These inputs, involved in centrifugal direction selectivity, suggest different circuit mechanisms underlying directional signal processing by distinct SAC subtypes [14].

of China (2020YFA0112700 to JH), grants STI2030-Major Projects(2021ZD0204500 to JH), Shanghai Municipal Science and Technology Major Project (2018SHZDZX05 to JH), the Strategic Priority Research Program of the Chinese Academy of Sciences (XDB32000000 to JH), the National Natural Science Foundation of China (31871035 to JH), and the State Key Laboratory of Neuroscience(to JH). The funders had no role in study design, data collection and analysis, decision to publish, or preparation of the manuscript.

**Competing interests:** The authors have declared that no competing interests exist.

**Abbreviations:** AC, amacrine cell; BAC, bacterial artificial chromosome; BP, bipolar cell; ChAT, choline acetyltransferase; dpf, days post-fertilization; DSI, direction selectivity index; dsRGC, direction-selective retinal ganglion cell; hpf, hours post-fertilization; INL, inner nuclear layer; IPL, inner plexiform layer; LPT, line printer terminal; MC, müller cell; NPY, neuropeptide Y; OKR, optokinetic reflex; PD, preferential direction; ROI, region of interest; SAC, starburst amacrine cell; sgRNA, small guide RNA; TF, transcription factor.

GABAergic/cholinergic amacrine cells (ACs) were found in nearly all vertebrates from fish to humans [15–18]. In the mammalian retina, SACs are the only cholinergic retinal cells [19]. However, previous cross-species studies have also implied more than a single cholinergic type in some vertebrate species. The goldfish retina contains 4 populations of anti-choline acetyltransferase (ChAT) immunoreactive neurons, 2 in the INL, and 2 in the GCL, while the zebrafish retina has 4 ChAT-positive laminae in the inner plexiform layer (IPL) [17,18]. In the turtle and chicken retina, 3 populations of cholinergic ACs have been reported [20,21]. However, genetically distinct GABAergic/cholinergic AC types have not been systematically examined in the vertebrate retina.

In the vertebrate retina, *ptf1α* is a critical factor in specifying ACs [22–24], which are a diverse cell type [25,26]. In zebrafish, ACs have been classified into as many as 29 types according to cell size, soma's location, and neurite arborization, including narrow-field ACs (15 types), medium-field ACs (5 types), and wide-field ACs (7 types) [27,28]. Nevertheless, the majority of ACs of these types belong to 2 neurotransmitter types, either GABAergic or glycinergic [29]; however, there are a few ACs that release acetylcholine [17], dopamine [30,31], and neuromodulators neuropeptide Y (NPY) [32]. GABAergic ACs are characterized by monostratified dendritic arbors with some overlapping and are born earlier than glycinergic ACs [33–35]. Cholinergic ACs are a subpopulation of GABAergic ACs [4,36]. However, the cellular diversity of cholinergic ACs in the zebrafish remains unexplored.

In this study, combining single-cell transcriptomic data analysis, *in situ* hybridization, and transgenic fish lines, we identified 2 types of GABAergic/cholinergic ACs, *bhlhe22*+ type and *sox2*+ type. Additionally, we performed gene disruption using CRISPR/Cas9 and found that codisruption of *bhlhe22* and *bhlhe23* could largely eliminate *bhlhe22*+ type, while disruption of *sox2* led to the depletion of *sox2*+ type. The morphological analysis of 2 AC types showed that they were distinct in dendritic arbor size, dendritic lamination, and soma position. Furthermore, *in vivo* two-photon calcium imaging suggested directional selectivity for both AC types. Consistently, animals with either AC type ablation exhibited compromised saccade frequencies in response to directional moving gratings by optokinetic reflex (OKR) assay. Interestingly, we also found that lamina-defined ON and OFF *sox2*+ type switched response polarity and responded to both ON and OFF illumination. Finally, we systematically defined the difference in response properties of 2 AC types in detecting different-sized moving bars and found that compared to *sox2*+ type, *bhlhe22*+ type responded more robustly to larger-size objects, better response dynamics to smaller-size objects, and higher direction selectivity to a broader size range. Our study demonstrated the evolutionarily conserved *sox2*+ AC type as well as a previously undescribed *bhlhe22*+ AC type, providing the new knowledge of the cellular diversity of GABAergic/cholinergic ACs in the vertebrate retina.

## Results

### Single-cell RNA-seq reveals 2 transcriptome-defined GABAergic/cholinergic AC types

With an attempt to dissect the diversity of GABAergic/cholinergic ACs, we sought for ACs coexpressing *gad1b* (GABAergic) and *vachta* (cholinergic) [37] using single-cell transcriptome data of 72 hours post-fertilization (hpf) zebrafish retina that we obtained previously (S1A Fig) [38]. At 72 hpf, the differentiation of major cell types in the retina is thought to be mostly completed in zebrafish [38]. From the 19 retinal clusters, we identified 5 AC clusters based on coexpression of *elval3* and *tfap2a* (Figs 1A, 1B and S1B; see S1 Data). These 5 AC clusters could be readily differentiated by the expression of marker genes (AC1: *sox4a*; AC2: *gad2*; AC3: *glyt1*; AC4: *bhlhe22;* AC5: *sox2*). Notably, 2 AC clusters, AC4 and AC5, were coexpressing *gad1b* and *vachta*. Cells in AC4 were expressing transcription factors (TFs), *bhlhe22*, *bhlhe23*, and *meis2b*

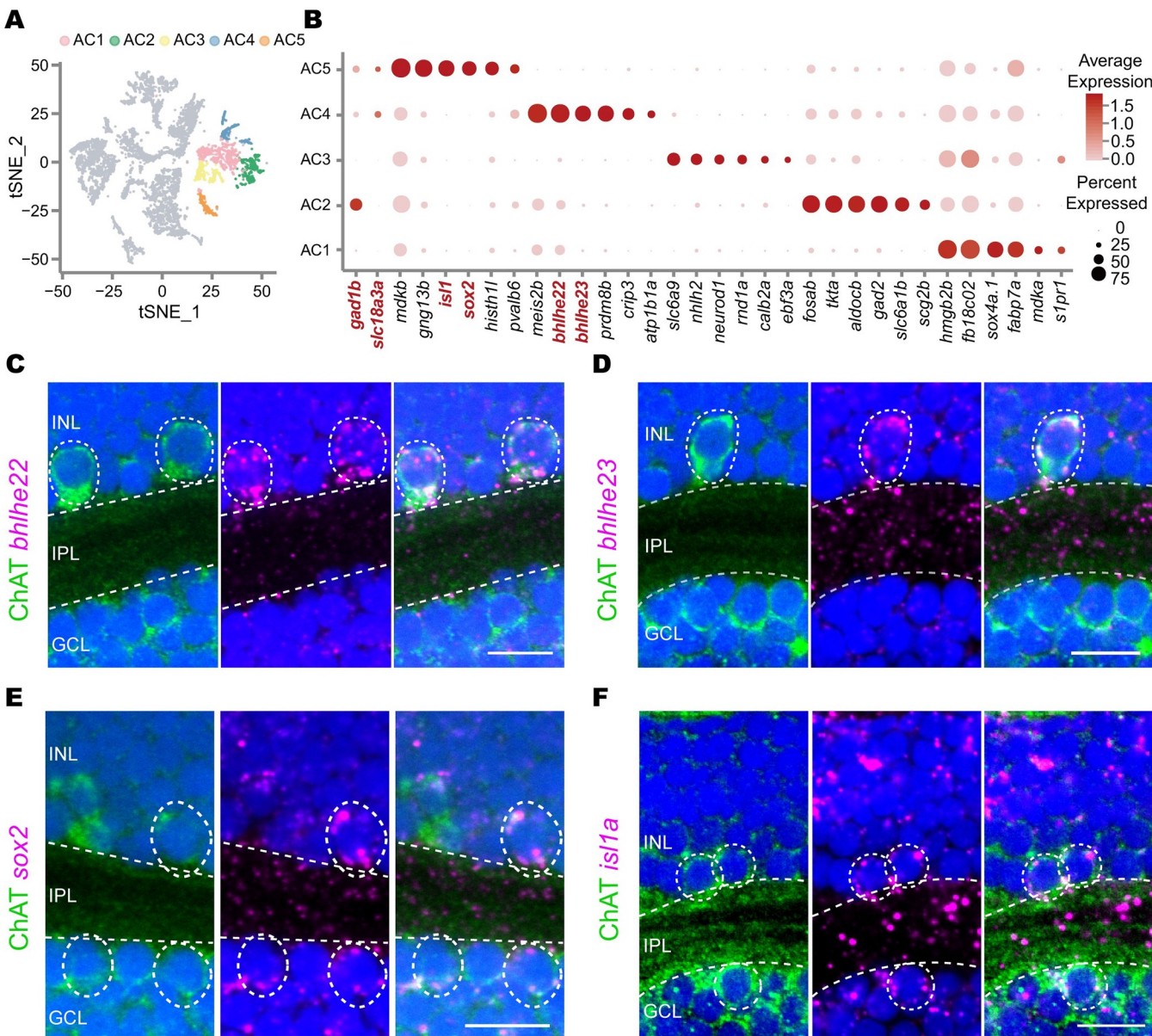

**Fig 1. Single-cell RNA-seq identifies 2 GABAergic/cholinergic AC types.** (**A**) t-SNE plot showing 5 clusters of ACs (in colors) in single-cell RNA-seq data. (**B**) Expression pattern of marker genes for each cluster in (**A**). Marker genes *gad1b* (GABAergic), *vachta* (cholinergic), *sox2* (AC5), *bhlhe22* (AC4), *glyt1* (AC3), *gad2* (AC2), and *sox4a* (AC1) are highlighted in red. (**C-F**) Validation of marker TFs for cluster AC4 (**C** and **D**) and AC5 (**E** and **F**) using *in situ* hybridization (magenta) combined with ChAT immunostaining (green) of the 5-dpf zebrafish. Dashed circles, ACs with positive signals. Scale bars, 10 μm. AC, amacrine cell; ChAT, choline acetyltransferase; dpf, days post-fertilization; GCL, ganglion cell layer; INL, inner nuclear layer; IPL, inner plexiform layer; TF, transcription factor.

(*bhlhe22*+ type), whereas those in AC5 expressed TFs, *sox2*, and *isl1a* (*sox2*+ type; Fig 1B). Note that *bhlhe22* was most abundantly expressed in AC4; *bhlhe23* was highly expressed in AC4 and bipolar cells (BPs); *sox2* was most abundantly expressed in AC5 and moderately expressed in müller cells (MCs); *isl1a* was highly expressed in AC5, RGCs, some BPs and horizontal cells (S1B–S1D Fig). Furthermore, *in situ* hybridization of *bhlhe22*, *bhlhe23*, *sox2*, and *isl1a* together with ChAT immunostaining showed that all these 4 genes were expressed in ACs that were ChAT-positive (Figs 1C–1F, S1F, and S1G). Also, the bodies of *bhlhe22*+ and

*bhlhe23*+ type were in the INL, whereas *sox2*+ and *isl1a*+ type resided in both the INL and GCL (Fig 1C–1F). Together, our analyses identified 2 types of GABAergic/cholinergic ACs, *bhlhe22*+ type and *sox2*+ type, in the zebrafish retina.

## Genetically marking 2 GABAergic/cholinergic AC types

To genetically mark 2 AC types, we generated 3 transgenic lines that specifically identify neurons expressing *bhlhe22*, *bhlhe23*, or *sox2* with mNeonGreen or mRuby3 using bacterial artificial chromosome (BAC) recombination (Figs 2A–2D, 2G, 2H, S4A and S4B) [39]. We found that TgBAC(*bhlhe22*: *mNeonGreen*) could specifically mark a subset of ChAT+ and Gad65/67 + ACs located in the INL (Fig 2A, 2J and 2L). Together with the *in situ* result above (Fig 1C), our results indicated that TgBAC(*bhlhe22*: *mNeonGreen*) specifically marked *bhlhe22*+ type in the retina. In addition, TgBAC(*bhlhe23*: *mRuby3*) specifically marked *bhlhe22*+ type (98%) and a number of BPs (Fig 2E and 2F), which is consistent with the result of single-cell transcriptome analysis (S1C Fig). On the other hand, TgBAC(*sox2*: *mNeonGreen*) specifically marked a ChAT+ and Gad65/67+ subset of ACs located in the INL and the GCL (Fig 2G, 2K and 2M), which was consistent with single-cell transcriptome data (S1D Fig). We verified that cells marked by this transgenic line were colabeled with a SOX2 antibody using the immunostaining (Fig 2G and 2H). Thus, TgBAC(*sox2*: *mNeonGreen*) could specifically mark *sox2*+ type and MCs. Additionally, *in situ* hybridization of *isl1a* combined with SOX2 immunostaining showed that the majority of *sox2*+ type coexpressed with *isl1a* (S1H and S1I Fig). Furthermore, TgBAC(*sox2:mNeonGreen, bhlhe23:mRuby3*) showed that the 2 AC types were distributed in distinct but intermingled patterns (S2C–S2E Fig) and constituted a total of approximately 80% of cholinergic ACs in the retina (S2E Fig). Together, we created the transgenic lines specifically marking *bhlhe22*+ type and *sox2*+ type, which constituted a large population of GABAergic/cholinergic ACs in the zebrafish retina.

## Disruption of distinct sets of TFs influences the generation of 2 AC types

Next, we examined the roles of type-specific TFs (*bhlhe22*, *bhlhe23*, and *sox2*) in specifying 2 AC types by CRISPR/Cas9-mediated gene disruption in the founder generation (F0/G0 mutants) [40] (S2A and S2B Fig). In TgBAC(*sox2:mNeonGreen,bhlhe23:mRuby3*) F0/G0 mutants, we found that codisruption of *bhlhe22* and *bhlhe23* led to a significant decrease in the number of *bhlhe22*+ type (15,556 and 1,728 cells/mm$^2$ of control and *bhlhe22*/*bhlhe23*-disrupted group, respectively, $p = 0.0014$), without the obvious influence on the number of *sox2*+ type, whereas disruption of either *bhlhe22* or *bhlhe23* had little influence on the number of *bhlhe22*+ type and *sox2*+ type (Fig 3A–3G, 43,016 and 35,309 cells/mm$^2$ of control and *bhlhe22*/*bhlhe23*-disrupted group, respectively, $p = 0.9031$).

On the other hand, disruption of *sox2* resulted in a significant decrease in the number of *sox2*+ type (1,212 cells/mm$^2$ of *sox2*-disrupted group, $p < 0.001$), leaving little change in the number of *bhlhe22*+ type (Fig 3A–3G, 14,949 cells/mm$^2$ of *sox2*-disrupted group, $p > 0.99$). Additionally, the absence of *sox2* led to an increase of *sox2* expression in MCs without changing the number of MCs, which were identified by glutamine synthetase expression (S3A and S3B Fig). Together, these results indicated that *bhlhe22*/*bhlhe23* and *sox2* were responsible for specifying *bhlhe22*+ type and *sox2*+ type, respectively.

Blockage in specifying *bhlhe22*+ type or *sox2*+ type could not result in the fate respecification between the two, which raised a question as to where these cells go. To explore this, we examined the influence of disrupting *bhlhe22*/*bhlhe23* or *sox2* on the proportions of retinal major cell types using Spectrum of Fates (SoFa1) transgenic line [41], which all retinal cell types are differentially labeled by combination of fluorescent proteins. The result did not show

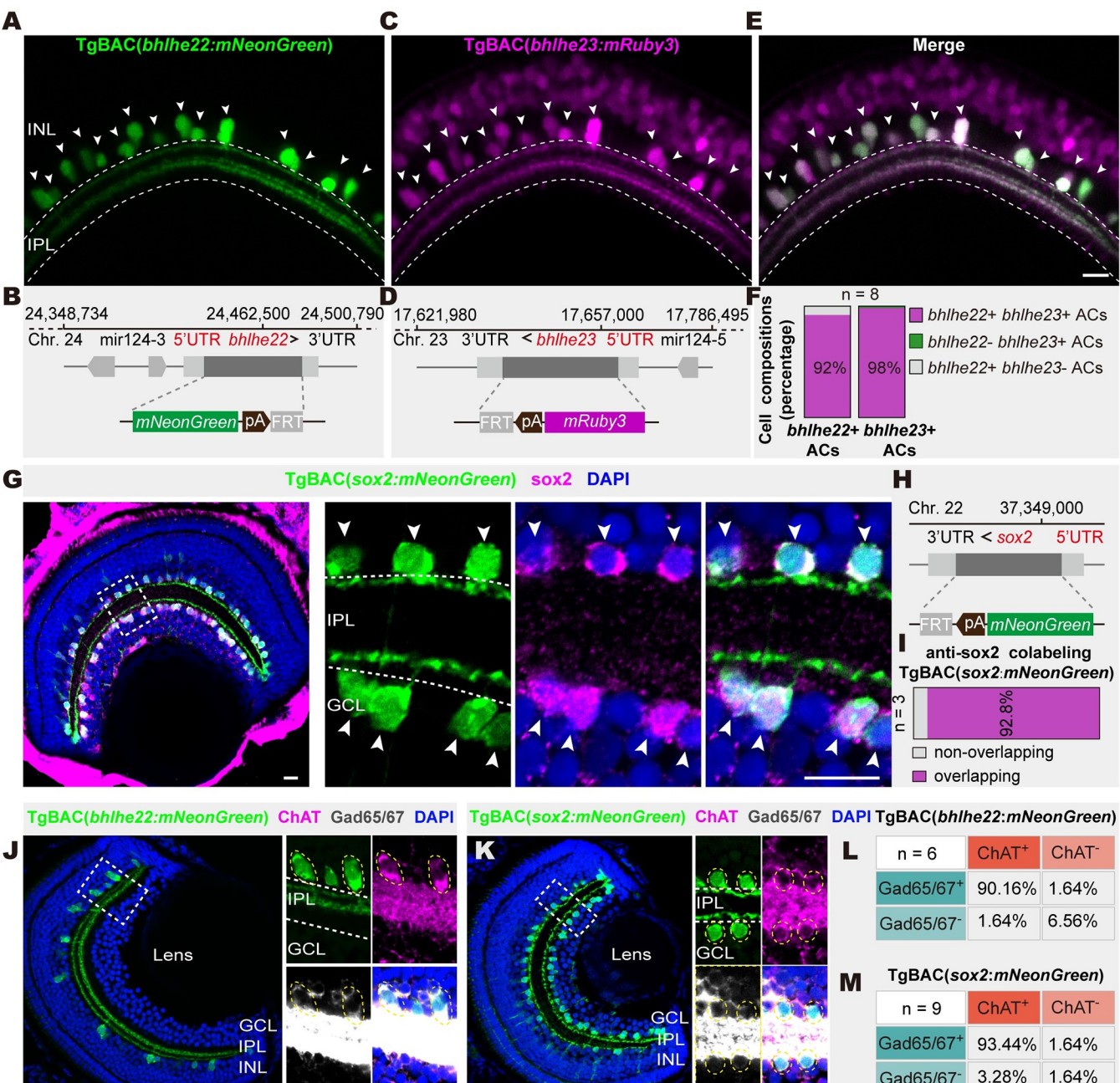

**Fig 2. Genetically marking 2 GABAergic/cholinergic AC types.** (**A** and **B**) Representative image (**A**) of derived transgenic line and schematic (**B**) of BAC construct design of *bhlhe22*. (**C** and **D**) Representative image (**C**) of derived transgenic line and schematic (**D**) of BAC construct design of *bhlhe23*. (**E**) Merged image of (**A**) and (**C**) after crossing transgenic lines TgBAC(*bhlhe22:mNeonGreen)* with TgBAC(*bhlhe23:mRuby3*). (**F**) Cell composition analysis of *bhlhe22* and *bhlhe23* labeling cells in (**E**). Larvae used for composition analysis are from TgBAC(*bhlhe22:mNeonGreen,bhlhe23:mRuby3*), n = 8. (**G**) Immunostaining of *sox2*+ ACs in the transgenic fishline TgBAC(*sox2:mNeonGreen*). Solid white arrow head indicated colocalization of sox2 transgenic fishline and SOX2 antibody. (**H**) Schematic of BAC construct design of TgBAC(*sox2: mNeonGreen*). (**I**) The bar plot showing that the majority of *sox2*+ ACs in the transgenic fishline is colocalized with SOX2 antibody. (**J** and **K**) Images showing the colabeling of *bhlhe22* + type (**J**, green) and *sox2*+ type (**K**, green) with ChAT (magenta) and Gad65/67 (gray). (**L** and **M**) Quantification in (**J** and **K**). Images above are captured from 5-dpf larval fish. Dashed yellow circles, ACs with positive signals. The data underlying this figure can be found in S3 Data. Scale bars, 10 µm. AC, amacrine cell; BAC, bacterial artificial chromosome; ChAT, choline acetyltransferase; dpf, days post-fertilization; GCL, ganglion cell layer; INL, inner nuclear layer; IPL, inner plexiform layer.

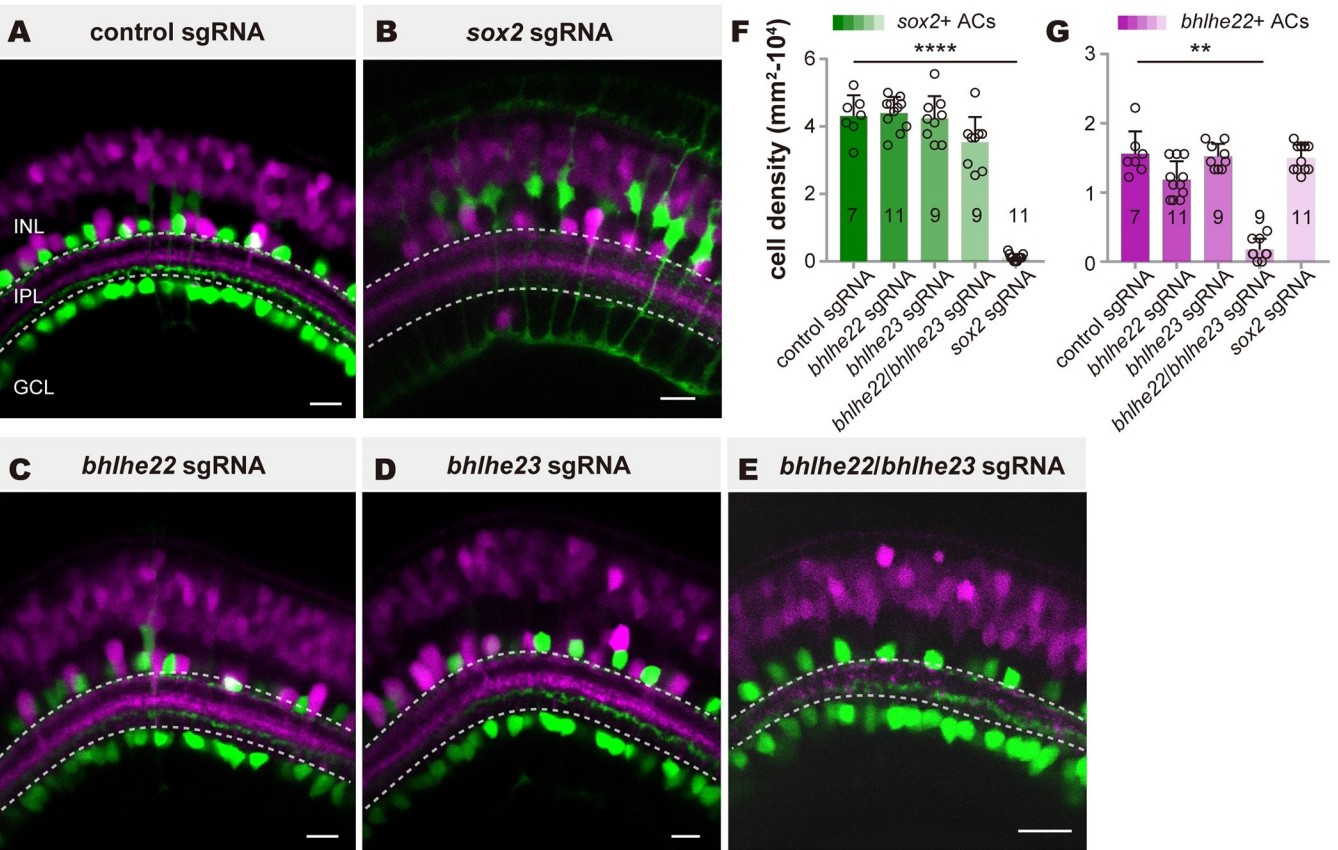

**Fig 3. The generation of 2 AC types requires distinct sets of TFs.** (**A-E**) Representative images showing the distributions of 2 GABAergic/cholinergic ACs labeling by TgBAC(*sox2*: *mNeonGreen,bhlhe23:mRuby3*) after injection of control sgRNA(**A**), *bhlhe22* sgRNA (**B**, *bhlhe22* disruption), *bhlhe23* sgRNA (**C**, *bhlhe23* disruption), *bhlhe22/bhlhe23* sgRNA (**D**, *bhlhe22* and *bhlhe23* codisruption), *sox2* sgRNA (**E**, *sox2* disruption) at 5 dpf. (**F**) Quantifications of *sox2+* type in (**A-E**). (**G**) Quantifications of *bhlhe22* + type in (**A-E**). The data underlying this figure can be found in S3 Data. Data are presented as mean ± SD, Mann–Whitney test. * $p < 0.05$, ** $p < 0.01$, *** $p < 0.001$, **** $p < 0.0001$. Scale bars, 10 μm. AC, amacrine cell; dpf, days post-fertilization; GCL, ganglion cell layer; INL, inner nuclear layer; IPL, inner plexiform layer; sgRNA, small guide RNA; TF, transcription factor.

any notable change in major retinal cell-type composition at the population level after the disruption of either *bhlhe22/bhlhe23* or *sox2* (S3C and S3D Fig). However, we could not rule out the possibility that SoFa1 lacked the sensitivity to detect changes of retinal subtypes. Also, cell death could be involved in the loss of 2 AC types after gene disruption.

## Two AC types are morphologically distinct

Furthermore, we examined cell morphology by analyzing single photo-converted GABAergic/cholinergic ACs, which were sparsely marked using newly generated lines TgBAC(*bhlhe23*: *gal4, uas:kaede*) and TgBAC(*sox2:gal4ff,uas:kaede*) (S4A and S4B Fig). Despite that both types exhibited symmetric dendritic arbors, the dendrites of *bhlhe22+* type were unbranched, and *sox2+* AC dendrites were branched in a starburst-like pattern reminiscent of mouse SACs [16]. More strikingly, the dendrites of *bhlhe22+* type (ON-laminae subtype: 81.39 ± 20.14 μm in diameter and 3.22 ± 1.98 × $10^3$ μm$^2$ in size, $n = 20$; OFF-laminae subtype: 94.73 ± 8.08 μm in diameter and 4.24 ± 1.33 × $10^3$ μm$^2$ in size, $n = 3$) were about 2.9 times larger than that of *sox2+* type (ON-laminae subtype: 27.95 ± 4.13 μm in diameter and 0.36 ± 0.09 × $10^3$ μm$^2$ in size, $n = 13$; OFF-laminae subtype: 31.63 ± 4.67 μm in diameter and 0.42 ± 0.10 × $10^3$ μm$^2$ in size, $n = 18$) (Figs 4A, 4C, 4D, S4C and S8C) [16,42].

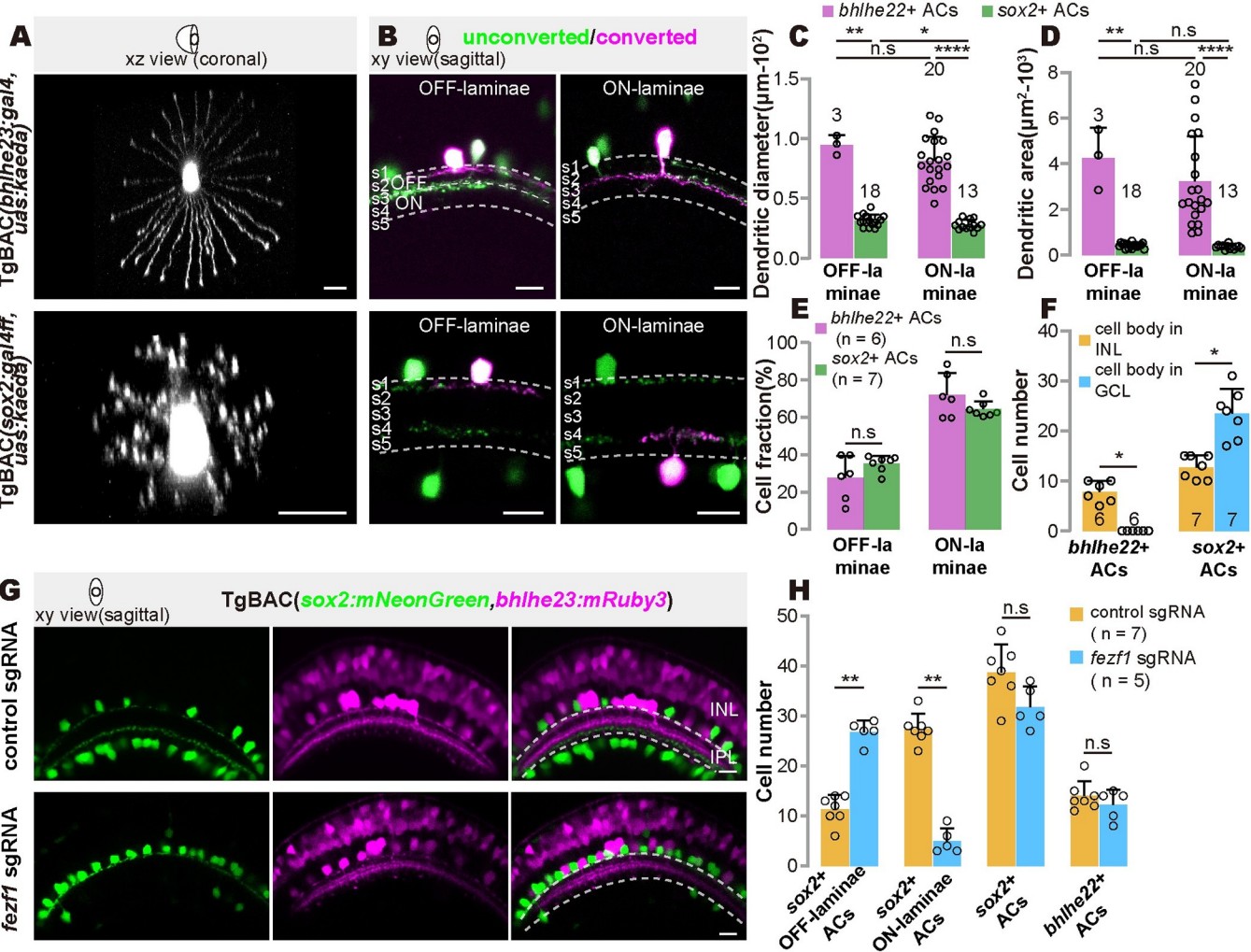

**Fig 4. Morphological characteristics of 2 GABAergic/cholinergic AC types.** (**A**) Representative images showing the dendrites of *bhlhe22*+ type and *sox2*+ type. (**B**) Representative images showing the cell body positioning and dendritic arbors of ON- and OFF-laminae of *bhlhe22*+ type and *sox2*+ type. (**C** and **D**) Quantification of the dendrite diameters (**C**) and sizes (**D**) in (**A**). Each circle represents one cell at 5 dpf. (**E**) Quantification of cell fraction of ON- to OFF-laminae in (**B**). (**F**) Quantification of the number of cells with cell body located in INL and GCL of *bhlhe22*+ and *sox2*+ type. (**G**) Representative images of *bhlhe22*+ and *sox2*+ type after *fezf1* disruption. (**H**) Quantification of the number of ON- and OFF-laminae subtype of *sox2*+ cells, total *sox2*+ cells, and total *bhlhe22*+ cells in (**G**). Each circle represents one fish at 5 dpf. The data underlying this figure can be found in S3 Data. Scale bars, 10 μm. AC, amacrine cell; dpf, days post-fertilization; GCL, ganglion cell layer; INL, inner nuclear layer; sgRNA, small guide RNA.

Both AC types comprised ON- and OFF-laminae subtypes. *Sox2*+ type consisted of ON-(*s4/s5* lamina) and OFF-laminae (*s1* lamina) subtypes, whose cell bodies were in the GCL and INL, respectively, reminiscent of mammalian SACs (Figs 4B, 4E, 4F, S8A and S8B) [13]. A recent study of the mouse retina elegantly showed that *fezf1* was required for positioning ON-laminae subtype SACs in the GCL [11]. We then examined the expression of *fezf1* in 2 AC types in single-cell transcriptome data and found its specific expression in *sox2*+ type (S1E Fig). Further genetic disruption of *fezf1* resulted in the positioning of the soma of *sox2*+ type from the GCL to the INL, with an increased number of OFF-laminae *sox2*+ type in the INLs, a decreased number of ON-laminae *sox2*+ type in the GCLs but no change in total *sox2*+ ACs and total *bhlhe22*+ ACs (Fig 4G and 4H). All these results supported an evolutionary counterpart relationship between zebrafish *sox2*+ type and mouse SACs (S8 Fig) [11]. In contrast,

although *bhlhe22*+ type comprised ON- (*s3/s4* lamina) and OFF-laminae (*s1* lamina) subtypes, ON-laminae subtypes' soma was in the INL rather than being displaced in the GCL (Figs 4B, 4E, 4F, S8A and S8B). Based on the early study of the morphological description of ACs in the zebrafish retina [27], *bhlhe22*+ type and *sox2*+ type are likely wide- and narrow-field ACs, respectively.

Thus, 2 GABAergic/cholinergic AC types exhibited distinct morphological characteristics in the dendrite branching, arbor size, and lamination.

## A small fraction of 2 AC types showed moderate directional responses

To examine whether 2 AC types exhibited direction selectivity, we performed *in vivo* two-photon calcium imaging of single *bhlhe22*+ or *sox2*+ AC to full-field moving bars at 12 directions (spanning 0˚ to 360˚ at 30˚ intervals) in paralyzed larval fish (5 to 8 days post-fertilization (dpf)) [43]. *bhlhe22*+ or *sox2*+ AC was sparsely marked by injecting *itol2-14uas*: *sypb-gcamp6s* plasmid into TgBAC(*bhlhe22*: *gal4ff*) and TgBAC(*sox2:gal4ff*), respectively (Fig 5A).

We collected a total of 1,208 and 1,163 regions of interests (ROIs; 1.9 × 1.9 μm, local dendrites; ROI selection, see Materials and methods, Fig 5B and 5F) from *bhlhe22*+ type (27 animals) and *sox2*+ type (25 animals), respectively. Around 93.6% of *bhlhe22*+ ROIs and 88.2% of *sox2*+ ROIs showed qualified responses to moving bars (Fig 5C and 5G). Overall, the ROI response amplitude at local dendrites of *sox2*+ type was significantly larger than those of *bhlhe22*+ type (S5C Fig, dF/F0: 0.72 ± 0.58 and 1.33 ± 1.27 for *bhlhe22*+ type and *sox2*+ type, respectively; $p < 0.0001$). To determine direction-selective responses, we calculated direction selectivity index (DSI) based on ROIs' responses to moving bars at 12 directions. Direction-selective property of local dendrites was then determined based on DSI value (DSI > = 0.5; see Materials and methods). Surprisingly, unlike mouse SACs that uniformly show robust directional responses [44], we found that 7.8% (*bhlhe22*+ type, $n = 94$) and 28.8% (*sox2*+ type, $n = 335$) of total ROIs showed direction selectivity (Fig 5C and 5G). Direction-selective *bhlhe22*+ and *sox2*+ ROIs showed higher responses at PDs than null directions (Fig 5D, 5E, 5H, 5I, S5A and S5B), and the DSI value of pooled direction-selective ROIs (*bhlhe22*+ and *sox2*+ ROIs) was statistically higher than those of non-direction-selective ROIs (S5G Fig). Additionally, the responses and PD of direction-selective ROIs of both AC types could occur at all 4 quadrants, although distribution of PD of 2 types of ROIs was different (S5E and S5H Fig). Moreover, the direction tunning capacity (depicted by DSI value) of *sox2*+ ROIs was significantly higher than *bhlhe22*+ ROIs (Fig 5J and 5K). However, due to the fact that the small proportion of both AC types showed moderate directional responses, their direction-selective properties are needed to be further determined by other methods in the future.

## Genetic ablation of either AC type compromised optokinetic reflex

OKR is a behavior that relies on the activity of retinal direction-selective neurons, such as SACs in the mouse retina [44,45]. To further evaluate the function of 2 AC types in directional selectivity at the behavioral level, we examined the OKR in response to drifting gratings in approximately 7- to 8-dpf larval zebrafish with the ablation of either AC type (Fig 5L and 5M). We ablated *bhlhe22*+ and *sox2*+ type by *bhlhe22/bhlhe23* double disruption and the nitroreductase/metronidazole system, respectively (S6A and S6B Fig). We determined the optimal parameters of drifting gratings that triggered the OKR (contrast: 1.0; spatial resolution: 0.026 cycle/degree). In response to this optimal stimulation, both *bhlhe22*+ type- and *sox2*+ type-ablated zebrafish exhibited a significant reduction in saccade frequency (Fig 5N), suggesting a potential role of both AC types in detecting directionally moving objects. However, the saccade frequency of the zebrafish with the ablation of either *bhlhe22* or *sox2* was not completely

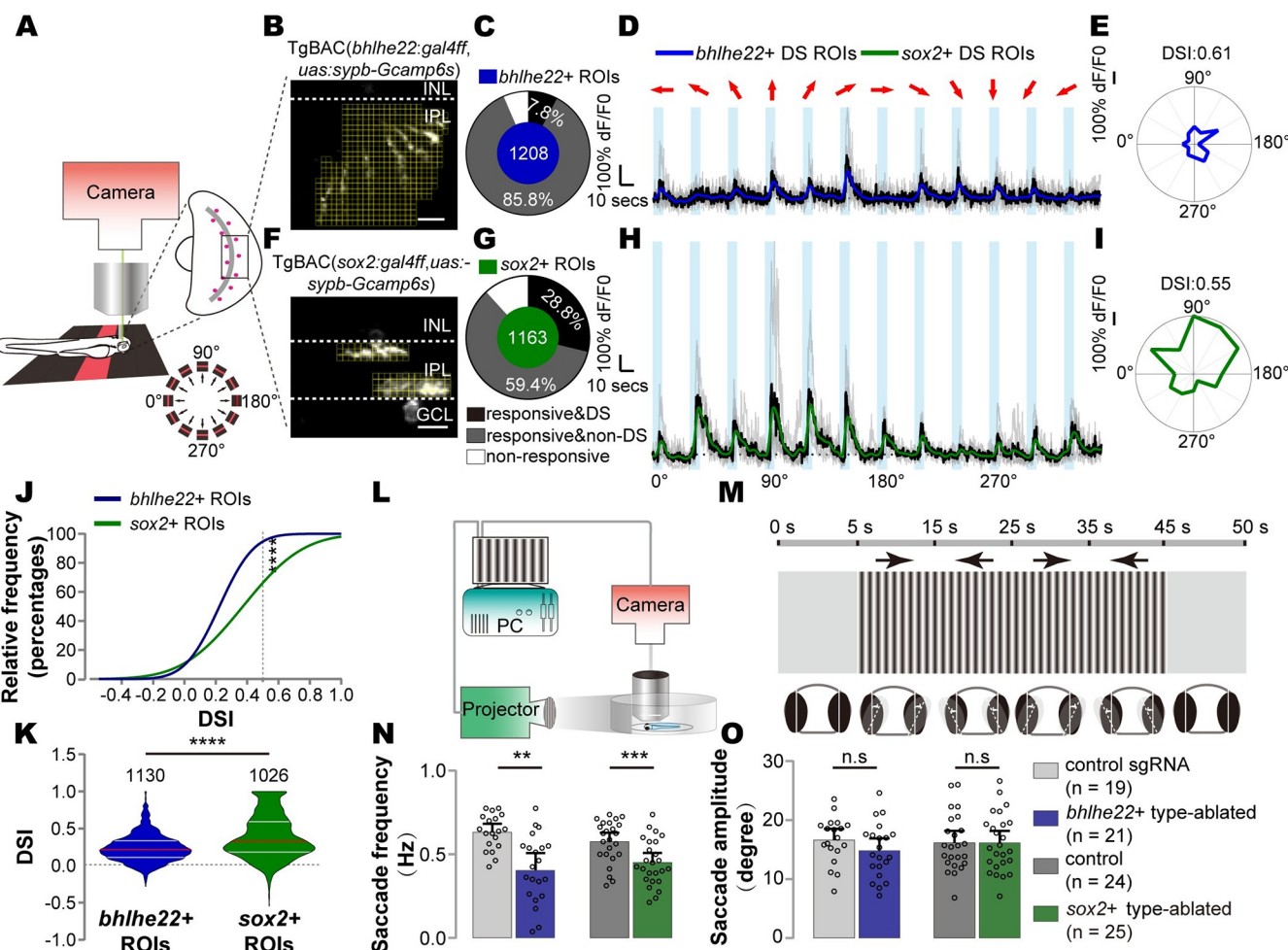

**Fig 5. Direction selectivity of 2 GABAergic/cholinergic AC types.** (**A**) Schematic showing the setup of *in vivo* two-photon calcium imaging of 5- to 8-dpf TgBAC(*sox2*: *gal4ff,uas:sypb-gcamp6s*) and TgBAC(*bhlhe22:gal4ff,uas:sypb-gcamp6s*) fish. (**B**) The representative image of TgBAC(*bhlhe22: gal4ff,uas:sypb-gcamp6s*) fish. Dendrites in the section are segmented in tiling yellow rectangle. A single yellow rectangle is an ROI for calcium response analysis. (**C**) Donut plot showing the composition of responsive and direction-selective *bhlhe22*+ ROIs to 50-pixels width moving bar. (**D**) Responses of a representative ROI of DS *bhlhe22*+ ROIs in (**C**). Gray traces, repetitively trials; black traces, average of repetitive trials; blue trace, smoothed data of average trace with gaussian method. (**E**) Polar plot showing the average response at 12 directions of the ROI in (**D**). (**F**) The representative image of TgBAC(*sox2*: *gal4ff,uas:sypb-gcamp6s*) fish. (**G**) Donut plot showing the composition of responsive and direction-selective *sox2*+ ROIs. (**H**) Responses of a representative ROI of DS *sox2*+ ROIs in (**G**). Green trace, smoothed data of average trace with gaussian method. (**I**) Polar plot showing the average response in 12 directions of the ROI in (**H**). (**J**) Cumulative curve of DSI distribution of responsive *sox2*+ (green) and *bhlhe22*+ (blue) ROIs. The dashed line indicates the direction-selective threshold, DSI = 0.5. Kolmogorov–Smirnov test. (**K**) Violin plot of DSI distribution of all responsive *sox2*+ and *bhlhe22*+ ROIs. The gray lines indicate the quartiles, and the red lines indicate the median. (**L** and **M**) Behavioral paradigm for the OKR assay. (**N**) The bar plot showing saccade frequency of *bhlhe22*+ AC-ablated fish (magenta) and *sox2*+ AC-ablated fish (green) in response to drifting gratings. Each circle represented one fish at 5–8 dpf. (**O**) The bar plot showing saccade amplitude of *bhlhe22*+ AC-ablated fish (magenta) and *sox2*+ AC-ablated fish (green) in response to drifting gratings. Each circle represented one fish at 5–8 dpf. The data underlying this figure can be found in S3 Data. Data are presented as mean ± 95% CI, Mann–Whitney test, n.s > 0.005, ** *p* < = 0.005, *** *p* < 0.001, **** *p* < 0.0001. AC, amacrine cell; dpf, days post-fertilization; DS, direction selective; DSI, direction selectivity index; OKR, optokinetic reflex; ROI, region of interest; sgRNA, small guide RNA.

eliminated, which was in consistent with the finding of the small fraction of 2 AC types ROIs with DS characteristics.

To evaluate the influence of ablating other *bhlhe22*- and *sox2*-expressing cells in the retina or other brain regions on the OKR defects, we first examined retinal cells and found that the cell ablation had little influence on the numbers of neighboring retinal cells, including MCs (S6C–S6J Fig). Also, we did not observe the expression of *bhlhe22* in the brain of TgBAC

(*bhlhe22*: *mNeoGreen*) but did observe the expression of *sox2* in a small portion of neurons in the telencephalon and optic tectum, which was consistent with previous studies [46] (S1–S3 Movies). Furthermore, we found that the saccade amplitude was not significantly affected in cell-ablated animals (Fig 5O and S4–S7 Movies), indicating no motor defect. All these results supported a major contribution of ablating *bhlhe22*+ or *sox2*+ type in the retina to OKR defects. However, we could not completely rule out the potential influence of small subpopulations of *sox2*-expressing neurons in the telencephalon and the optic tectum because neurons in both regions, which showed the activity correlated with the OKR [47].

## ON/OFF-responsive AC subtypes show little lamina specificity

On the other hand, we performed *in vivo* two-photon calcium imaging of single *bhlhe22*+ or *sox2*+ ROIs responding to light flash (a red spot) to determine ON and OFF responses. Among 1,145 responsive *bhlhe22*+ ROIs, 842 were ON- and 303 were OFF-responsive neurons, whereas among 880 responsive *sox2*+ ROIs, 431 showed ON responses and 449 showed OFF responses (S5I–S5M Fig).

Previous studies proposed that ACs stratified in the outer half (*s1-s2*) and inner half (*s3-s5*) of the IPL were associated with OFF and ON responses, respectively [48,49]. We directly examined whether ON and OFF response subtypes (for both AC types) showed lamina specificity. The location of *sox2*+ soma could be used to determine lamina-defined ON and OFF subtypes. Interestingly, regardless of ON and OFF responses, *sox2*+ type could project their dendrites at both ON-laminae (*s3-s5*) and OFF-laminae (*s1-s2*) in the IPL (S5M Fig). This result suggested that the dendrites of ON and OFF response *sox2*+ type scarcely showed lamina specificity, arguing for the idea of the assignment of ON and OFF response properties of ACs by the lamina positions of their dendrites, which is also suggested by the work on fish BPs [50] that dendritic stratification does not always correlate to ON and OFF responses in zebrafish.

In addition, we evaluated the response latency of 2 AC types using light-on stimuli as previously described [51]. We analyzed the response latency of all responsive trials (4 repetitive trials for each ROI) based on light flashes. The result showed that the time duration of *sox2* + ROIs was not statistically different with that of *bhlhe22*+ ROIs (1.74 ± 0.50 and 1.79 ± 0.50 seconds; S5D Fig).

Together, *sox2*+ AC type exhibited little correlation between laminar position and ON/OFF responsiveness at the single-cell level.

## *Bhlhe22*+ type and *sox2*+ type show preferential responses to different-sized objects

The previous study has showed that the soma of ACs with larger dendritic fields generated larger responses than those with smaller dendritic fields to the same size objects [52], suggesting that larger dendritic arbors gave rise to larger receptive fields, and thereby leading to more robust responses. Thus, the arbor size of ACs provides a reasonable measure to estimate AC's receptive field. Considering that *bhlhe22*+ ACs had the significantly larger dendritic arbors compared to *sox2*+ ACs, we hypothesized that *bhlhe22*+ ACs had larger receptive field than *sox2*+ ACs, thereby preferring different sizes of moving objects. In our study, we estimate [53] the receptive field by peak time and rising time (S7C–S7E Fig) of 2 AC types by smoothing data from pooling all the responsive trials (see Materials and methods). The result supported the idea that the receptive field of *bhlhe22*+ ACs was larger than *sox2*+ ACs.

To test this, we performed *in vivo* two-photon calcium imaging of *bhlhe22*+ type and *sox2*+ type in response to full-field moving bars at 12 directions (spanning 0˚ to 360˚ at 30˚ intervals) with 2 ranges of bar width (the small size range: 2, 5, 10 pixels; the large size range: 10, 50, 250

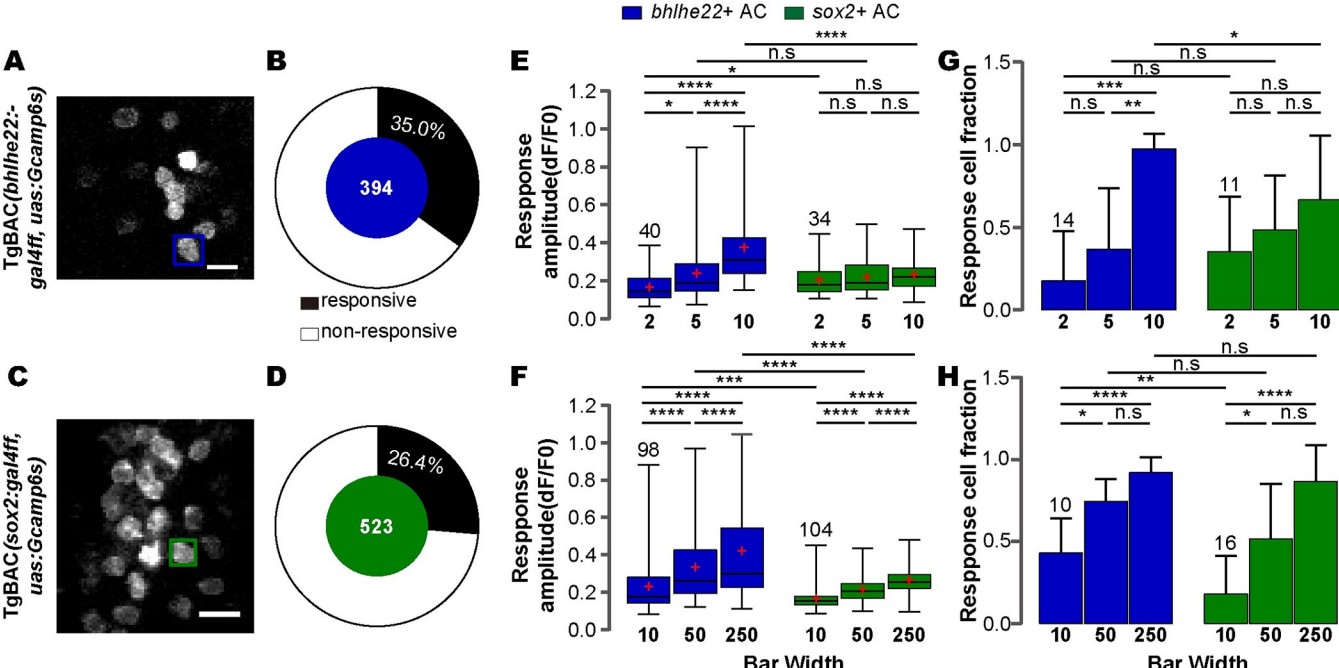

**Fig 6. *bhlhe22*+ and *sox2*+ type show conversely preference to object size.** (**A** and **B**) Representative capture (**A**) and donut plot (**B**) showing the responsive cell ratio of *bhlhe22*+ type to a small (2/5/10) and large (10/50/250) bar widths serial together. (**C** and **D**) Representative capture (**C**) and donut plot (**D**) showing the responsive cell ratio of *sox2*+ type to a small (2/5/10) and large (10/50/250) bar widths serial together. Rectangle on the capture showing an ROI of calcium activity analysis. (**E** and **F**) Bar plot showing the response amplitude of *bhlhe22*+ and *sox2*+ responsive type to small (**E**, *n* = 40 and *n* = 34 of *bhlhe22* + and *sox2*+ type soma, respectively) and large (**F**, *n* = 98 and *n* = 104 of *bhlhe22*+ and *sox2*+ type soma, respectively) bar widths serial. Friedman test and subsequent Dunn's multiple comparisons test within pairing wise groups (3 groups within small or large range). Mann–Whitney test between *bhlhe22*+ and *sox2*+ type groups. (**G** and **H**) Bar plot showing the responsive cell fraction of *bhlhe22*+ and *sox2*+ responsive type from TgBAC(*bhlhe22: gal4ff,uas:gcamp6s*) and TgBAC(*sox2:gal4ff,uas:gcamp6s*) to small (**G,** *n* = 14 and *n* = 11 of *bhlhe22*+ and *sox2*+ type animal, respectively) and large (**H,** *n* = 10 and *n* = 16 of *bhlhe22* + and *sox2*+ type animal, respectively) bar widths serial. The data underlying this figure can be found in S3 Data. Scale bars, 10 μm. n.s > = 0.05, * *p* < 0.05, ** *p* < 0.01, *** *p* < 0.001, **** *p* < 0.000.

pixels, 1°/pixel at visual distance of 0.36 cm, see Materials and methods). *Bhlhe22*+ type and *sox2*+ type were marked with GCAMP6s by crossing Tg(*uas*: *Gcamp6s*) with TgBAC(*bhlhe22*: *gal4ff*) and TgBAC(*sox2:gal4ff*), respectively (Fig 6A and 6C).

We analyzed calcium responses in the soma of 394 *bhlhe22* ACs and 523 *sox2*+ ACs to moving bars at 12 directions with either small or large range of bar width (the small size range, from 2 to 10 width units; the large size range, from 10 to 250 width units; see Materials and methods). In these cells, 35% *bhlhe22*+ ACs (138 out of 394 cells of 24 animals) and 26.4% *sox2* + ACs (138 out of 523 cells of 27 animals) showed the responses (Figs 6B, 6D, S7A and S7B). We first compared the responses of 2 AC types to moving bars of each width from the small to large size range. Notably, *sox2*+ ACs and *bhlhe22*+ ACs exhibited the preferential responses to moving bars of different sizes. Specifically, at the width of 2 (the smallest bar size we tested), *sox2*+ ACs showed higher response amplitude than those of *bhlhe22*+ ACs (dF/F0: 0.20 ± 0.08 (*sox2*+ type, *n* = 34) versus 0.17 ± 0.08 (*bhlhe22*+ type, *n* = 40), *p* = 0.022); at the width of 5, *sox2*+ ACs and *bhlhe22*+ ACs showed the statistically indistinguishable response amplitude (dF/F0: 0.22 ± 0.10 (*sox2*+ type, *n* = 34) versus 0.24 ± 0.16 (*bhlhe22*+ type, *n* = 40)) and responsive cell fraction (Fig 6E and 6G). However, at the width of 10 and larger, *bhlhe22*+ ACs showed the significantly higher response amplitude than *sox2*+ type (Fig 6F). Thus, *sox2*+ ACs showed more robust responses to the smallest size at the width of 2, whereas *bhhle22*+ ACs showed higher responses to the larger sizes at 10 or more.

Next, we evaluated the ability of either AC type (*bhhle22*+ type and *sox2*+ type) in distinguishing different sizes by examining the response change as a function of bar size change. Surprisingly, in response to an increasing width of moving bars throughout the small and large size ranges, *bhlhe22*+ ACs significantly elevated the response amplitude while increasing responsive cell fraction. In contrast, *sox2*+ ACs significantly elevated the response amplitude while increasing responsive cell fraction only within the large but not the small size range (Figs 6E–6H, S7A and S7B). Thus, while *sox2*+ type could distinguish different bar sizes within the large size range (10 to 250), *bhlhe22*+ type could distinguish the size difference within a broader range, from the small size range to the large size range (2 to 250).

Taken together, 2 AC types showed the differential responses to moving bars of different sizes. *sox2*+ ACs showed more robust responses at the smallest bar width of 2, and *bhlhe22* + ACs showed stronger responses to larger size bars at bar width of 10 and more. In terms of distinguishing different sizes, *bhlhe22*+ ACs, but not *sox2*+ ACs, could distinguish different bar sizes within the small size range, although both AC types could distinguish different bar sizes within the large size range.

## Discussion

Using single-cell RNA sequencing, we identified 2 types of GABAergic/cholinergic ACs in the zebrafish retina, *bhlhe22*+ type and *sox2*+ type. Codisruption of *bhlhe22* and *bhlhe23* resulted in a loss of *bhlhe22*+ type, while disruption of *sox2* led to a loss of *sox2*+ type. The 2 AC types exhibited distinct morphological characteristics in terms of dendritic arbor size, dendritic lamination, and soma position. Surprisingly, unlike mouse SACs, only a small fraction of 2 AC types showed moderate directional responses. Further analysis showed that compared to *sox2* + type, *bhlhe22*+ type gave rise to more robust responses to larger-size moving objects.

### Across-species comparison of cholinergic ACs

In the mouse retina, there is only one type of direction-selective GABAergic/cholinergic cells, known as SACs, which are expressing *sox2* [13]. SACs were also reported in the retina of other mammals, including cat, rabbit, macaque, baboon, and human [54–56]. Although cholinergic ACs are observed in many nonmammalian species, such as chick, turtle, and goldfish, it remains unknown whether they represent nonmammalian counterparts for mammalian SACs. In the current study, we identified a type of cholinergic ACs, *sox2*+ type. The transcription factor, *sox2*, was required for its fate specification. *sox2*+ type exhibited starburst-like branched symmetric dendrites. It comprised 2 subtypes, lamina-defined OFF subtype located in the INL and ON subtype in the GCL. Notably, *fezf1* disruption in *sox2*+ type led to lamina-defined ON subtype in GCL switching to OFF subtype in INL. However, in terms of directional responses, only a portion of *sox2*+ type showed directional responses, which is different from mouse starburst counterparts [12,57].

On the other hand, the finding of a new cholinergic *bhlhe22*+ type is intriguing. This new cholinergic type exhibited significantly larger dendritic arbors than *sox2*+ type. Functional assays further showed that *bhlhe22*+ type could detect large-size moving objects better than *sox2*+ type. It raises the possibility that the presence of *bhlhe22*+ type may provide advantages for fish to survive in the aquatic environment. More interestingly, the previous study in the mouse retina also reported the presence of a subset of ACs that were expressing *bhlhe22* but without ChAT expression [58]. However, its function remains unknown. Future functional studies of *bhlhe22*+ type in the retina of more species, including mammals and nonmammals, can provide deeper insights into the evolution of the cholinergic systems in the vertebrate retina.

### Polarity switch of lamina-defined ON and OFF *sox2*+ type

It is interesting to observe that lamina-defined ON and OFF *sox2*+ AC subtypes lacked the responses specific to ON and OFF illumination, respectively. A similar phenotype was also reported in the mouse retina, in which lamina-defined ON and OFF SACs could show polarity switch of light-induced responses. This switch was proposed to be the result of pre-synaptic BPs via surround circuits in the outer retina rather than lateral inhibition in the inner retina [59], which may be also at work for *sox2*+ type in zebrafish. Alternatively, either ON or OFF *sox2*+ type could receive inputs from both ON- and OFF-responsive BPs. A recent study showed that some ON BPs could make ectopic synapses in the OFF subla-mina [60]. Meanwhile, the lateral inhibition of neighboring ACs was reported to have a role of masking ON responses in OFF-RGCs, and gap junction could decrease this lateral inhibition, thereby resulting in polarity switch of OFF responses [61,62]. Besides, we could not rule out the possibility of the presence of undefined *sox2*+ AC subtypes. Future single-cell transcriptomic analysis of enriched sox2+ AC populations may provide the answer. In addition, it is also interesting to examine at the single-cell level if lamina-defined *bhlhe22*+ AC subtypes also lack specific ON or OFF responses. This requires future efforts to solve the technical difficulty of genetically marking lamina-defined ON and OFF *bhlhe22*+ type.

### Dendritic size and object size detection

It is interesting to observe that compared to *sox2*+ type, *bhlhe22*+ type with larger dendrites showed the advantage of distinguishing different bars within the small size range but not within the large size range. The first possibility is that larger-sized dendrites are more sensitive to detect size changes than smaller-sized ones. The previous study showed that the soma of ACs with larger dendritic fields generated larger responses than those with smaller dendritic fields to the same size objects [52]. Alternatively, it is possible that the larger dendrites of *bhlhe22*+ type are more densely tiling in the IPL than *sox2*+ type. Therefore, *bhlhe22*+ type within a unitary retina region could receive more upstream inputs from BPs [63,64], and neighboring *bhlhe22*+ type could also form more extensive crosstalk than *sox2*+ type via chem-ical or electrical coupling [65–67]. The more numerous upstream inputs and extensive cell crosstalk could enhance the response of the *bhlhe22*+ type as a function of bar size. Another possibility may involve the role of glycine regulation of *bhlhe22*+ type and *sox2*+ type. The pre-vious study showed that negative regulation of GABAergic inhibition by glycine could increase the edge detection of GABAergic ACs [68]. Thus, *bhlhe22*+ type with larger dendrites may receive more the glycinergic inhibition than *sox2*+ type with smaller dendrites resulting in an enhanced detection of different bar sizes within the small size range by *bhlhe22*+ type. The future investigation of all these possibilities can provide deeper insight into how these 2 newly identified GABAergic/cholinergic ACs in encoding retinal direction selectivity at the circuit level.

## Materials and methods

### Ethics statement

All animal procedures performed in this study were approved by Biological Research Ethics Committee of Center for Excellence in Brain Science and Intelligence Technology, Chinese Academy of Sciences with the approval number NA-045-2022. All the animal experiments were conducted based on the trial guideline on review of science and technology ethics, which was jointly released by the Ministry of Science and Technology, the Ministry of Education, the

Ministry of Industry and Information Technology, the National Health Commission, and other departments of the People's Republic of China.

## Zebrafish maintenance

Zebrafish were raised and maintained at 28°C, 12/12 hours light/dark cycles. Embryos were harvested and kept in the embryo medium (0.294 g/L NaCl, 0.0127 g/L KCl, 0.0485 g/L CaCl$_2$·2H$_2$O, 0.0813 g/L MgSO4·7H$_2$O, 0.3 g/L sea salt, and $2 \times 10^{-4}$ g/L methylene blue). Embryos were brought up in the embryo medium added with 0.003% phenylthiourea (PTU, Sigma-Aldrich) from 8 hpf to delay the pigmentation and anaesthetized by 0.04% ethyl 3-aminobenzoate methanesulfonate salt (MS-222, Sigma-Aldrich) before live imaging.

## Construct and transgenic lines

To construct 14uas: sypb-Gcamp6s, *synaptophysin* was cloned by from zebrafish genomic and recombined with plasmid 14uas: Gcamp6s. The published transgenic fishline used are Tg (*ptf1a*:*EGFP*) (jh1Tg) [69], Tg(*uas*:*kaede*) (s1999tTg) [70], and TgBAC(*slc6a9*:*EGFP*) (ion85Tg) [71]. Tg(*uas*:*Gcamp6s*) (nkUAShspzGCaMP6s13aTg) [72] was a gift from Dr. Jiulin Du. Sofa 1(cu2Tg; jh1Tg; q20Tg) transgenic fish is a gift from Dr. Harris [41]. The plasmid UAS:nfsB-mCherry was a gift from Dr. Toshio Ohshima (Waseda University, Tokyo, Japan) [73]. Tg(*uas*:*nfsB-mCherry*)(ion17hTg) was generated in lab. Transgenic lines were generated by BAC plasmid injection at one-cell stage. BAC plasmids (30 ng/μl) were coinjected with *tol2* mRNA (50 ng/μl). To generate TgBAC(*bhlhe22*:*mNeonGreen*) (ion10hTg), TgBAC(*bhlhe22*: *gal4ff*) (ion11hTg), TgBAC(*bhlhe23*:*mNeonGreen*) (ion12hTg), TgBAC(*bhlhe23*:*mRuby3*) (ion13hTg), TgBAC(*bhlhe23*:*gal4*) (ion14hTg), TgBAC(*sox2*:*mNeonGreen*) (ion15hTg), and TgBAC(*sox2*:*gal4ff*) (ion16hTg), original BAC clones CH211-277B21 (*bhlhe22*), CH211-237C5 (*bhlhe23*), and CH73-242D14 (*sox2*) were obtained from the BACPAC Resources Center and modified as the procedures described previously [39,74]. Primers used are listed in S2 Data.

## BAC construct recombination

The plasmids of PIGCN21-mNeonGreen-PA-frt-neo-frt and PIGCN21-gal4ff-PA-frt-neo-frt were constructed as DNA templates for BAC plasmid recombination. The cassette gal4ff-PA-frt-neo-frt or mNeonGreen-PA-frt-neo-frt was amplified from the plasmids with the primers (S2 Data) containing the homolog arms of *bhlhe22* or *sox2*. The plasmid PIGCN21-bhlhe23Arms-gal4-PA-frt-neo-frt, PIGCN21-bhlhe23Arms-mNeonGreen-PA-frt-neo-frt, and PIGCN21-bhlhe23Arms-mRuby3-PA-frt-neo-frt were constructed by multiple DNA fragments homologous recombination (Vazyme, China). A DNA fragment (1,470 bp) was amplified from the upstream of the start codon of *the bhlhe23* gene in CH211-237C5 and inserted into the upstream of the start codon of gal4 or mRuby3 as the left homolog arm (bhlhe23armL) in PIGCN21 vector. And, a DNA fragment (678 bp) was amplified from the downstream of the stop codon of *bhlhe23* and inserted into the downstream of frt-neo-frt as the right homolog arm (bhlhe23armR). The cassettes bhlhe23armL-gal4-PA-frt-neo-frt-bhlhe23armR, bhlhe23armL-mNeonGreen-PA-frt-neo-frt-bhlhe23armR, and bhlhe23armL-mRuby3-PA-frt-neo-frt-bhlhe23armR were amplified from the plasmids for BAC plasmid recombination. First, the iTol2-amp cassette was amplified from the piTol2-Amp plasmid and inserted into the original BAC clones by homologous recombination. The DNA fragment amplified from PIGCN21 plasmids was then recombined to replace the coding region of genes at the iTol2-amp BAC DNA in the sw105 bacterial strain. The neo selection marker was cut off by 0.1% L-arabinose (Sigma-Aldrich) at 32°C for 1 hour.

## Single-cell sequencing data analysis

Single-cell FASTQ raw data of 72-hpf retina was obtained from the previous study [38]. The raw data were converted to digital gene expression matrices after being mapped to the zebrafish genome (Zv10) using the Cell Ranger Single Cell Software Suite (version 2.1.0, 10×Genomics). For the first-round analysis, the Seurat R package (version 3.0) was applied [75]. Cells with less than 200 genes were filtered out. To ensure the clustering unbiasedly, cell-cycle-related genes were regressed. According to marker genes (S1 Data) of major cell types in the retina, total 19 clusters were identified. Then, all 5 clusters of ACs were selected for the further analysis. All differentially expressed genes in each AC cluster were identified. By the coexpression of cholinergic marker (*slc18a3a*) and GABAergic cell marker (*gad1b*), GABAergic/cholinergic ACs were defined. Ontology analysis was performed using R packages (clusterProfiler) and its related R packages.

## CRISPR-Cas9-based gene disruption

CRISPR-Cas9 system was applied to examine the function of the marker genes identified from the scRNA-seq data. Five genes, *bhlhe22*, *bhlhe23*, *sox2*, *isl1a*, and *fezf1*, were selected for further study. Four specific small guide RNA (sgRNA) sequences (S2 Data) of each gene were designed based on the coding sequences using the online tool CRISPRscan [76]. Four scrambled sgRNAs described in the previous study were used as control sgRNAs [40]. The plasmid pSQT1313(Addgene) containing the guide RNA scaffold was used as DNA template for the synthesis of guide RNA. A forward primer containing the T7 promoter, a gene-specific guide sequence, and a reverse primer on the guide RNA scaffold sequence were used to get PCR products for sgRNA transcription. The products were then transcribed to sgRNAs using T7 MEGAscript (Invitrogen). All sgRNAs were stored at −80˚C. Four sgRNAs for each gene were mixed (200 ng/μl) with Cas9 protein (400 ng/μl) (novoprotein) in RNase-free $H_2O$ right prior to the injection as described previously [40]. For *bhlhe23*/*bhlhe22* double knockout, 8 sgRNAs were mixed (final concentration of 200 ng/μl) with Cas9 protein (400 ng/μl) in $H_2O$.

## Tissue preparation and immunostaining

Embryos were harvested at 5 dpf. All samples were fixed in 4% paraformaldehyde (PFA, Sigma-Aldrich) immediately after they were anesthetized with 0.4% MS222 at 4˚C for about 8 hours. Samples were then cryoprotected in 30% sucrose for 6 hours, fresh-frozen, and cryosectioned at a thickness of 12 μm. Immunohistology details were described previously [5,6]. Sections were permeabilized in 1×PBS with 0.5% Triton X-100 for 30 minutes. After that, they were incubated with Quick Antigen Retrieval Solution for Frozen Sections (Beyotime, China) for 5 minutes and washed with Immunol Staining Wash Buffer (Beyotime, China). Then, they were blocked with 5% BSA solution (Sigma) at room temperature for 1 hour. Sections were incubated with the primary antibody at 4˚C overnight. The following day, sections were incubated with Alexa Fluor 488-, 568-, 594-, or 647-conjugated secondary antibodies. (1:1,000, Alexa Fluor 488 Donkey Anti-Goat IgG (H+L), Jackson ImmunoResearch, Cat#705-546-147; Alexa Fluor 568 Donkey anti-Goat IgG (H+L) Cross-Adsorbed Secondary Antibody, Invitrogen, Cat#A-11057; Alexa Fluor 594 Goat Anti-Rabbit IgG (H+L), Jackson ImmunoResearch, Cat#111-585-003; Alexa Fluor 647 Donkey Anti-Rabbit IgG (H+L), Jackson ImmunoResearch, Cat#711-605-152; Alexa Fluor 647 Donkey Anti-Goat IgG(H+L), Abcam, Cat# ab150131) at room temperature for 2 hours after washing out. For double immunostaining, sections were then blocked again after washing out of the secondary antibody. Then, the second primary antibodies were applied to sections and incubated at 4˚C overnight. Primary antibodies anti-ChAT antibody (1:100, host species: Goat, Cat#AB144P, lot number: 2464504, Merck), anti-

GAD65/GAD67 antibody (1:200, host species: Rabbit, Cat#AB11070, Abcam), and anti-sox2 antibody (1:200, host species: Rabbit, Cat#AB5603, Millipore) were used to detect cholinergic cells, GABAergic cells, and *sox2*-expressing cells, respectively.

## Riboprobe synthesis

Fluorescent *in situ* hybridization was performed to verify 4 specific markers (*bhlhe22*, *bhlhe23*, *sox2*, and *isl1a*) of GABAergic/cholinergic ACs. To prepare for RNA probe synthesis, total messenger RNA (mRNA) was extracted from the 3-dpf larvae by Trizol (Invitrogen). The total RNA was then reverse transcribed into complementary DNA (cDNA, TransGen,China). The cDNA was used as DNA template. A reverse primer (S2 Data) containing T7 promoter was used to obtain PCR products. After verifying the sequence of PCR products, they were then transcribed into antisense RNA probes via T7 RNA Polymerase kit (Promega) and RNA Labeling kit (Roche). Synthesized RNA probes are DIG-bhlhe23, DIG-sox2, DIG-*isl1a*, and FITC-bhlhe22 riboprobes.

## Fluorescent *in situ* hybridization

Samples were harvested at 5 dpf and fixed in 4% PFA at 4˚C overnight followed by the dehydration 30% sucrose. Thickness of frozen sections was 12 μm. To detect the riboprobe, antibody anti-DIG-POD (1:500, Roche, Cat#11207733910) or anti-FITC-POD (1:500, Roche, Cat#11426346910) was incubated at 4˚C overnight followed by TSA detection (PerkinElmer). Dilute TSA plus stock solution 1:100 in 1×amplification diluent to make TSA plus working solution. The fluorescence *in situ* hybridization was performed as described [74]. To check cell identity of riboprobe labeling cells, immunostaining was proceeded following TSA detection.

## Confocal image acquisition

Images were taken by an inverted confocal microscope system (FV1200, Olympus) using 60 × w/1.2 NA and 60 × s/1.3 NA objectives. All quantification and visualization were performed using FV10-ASW 4.2a Viewer (Olympus) and ImageJ (SPM12; http://www.fil.ion.ucl.ac.uk/spm/).

## Single-cell morphology analysis

Single GABAergic/cholinergic AC was selected from sparsely labelled TgBAC(*bhlhe23*: *gal4*, *uas*:*kaede*) or TgBAC(*sox2*:*gal4ff*,*uas*:*kaede*). It was photo-converted from kaede-green to kaede-red with the 405-nm laser. After 24 hours post-photoconversion, z-stack of images was acquired. To measure the dendritic size of cell, the x and y dimension of images were resized equivalently to z dimension (0.5 μm) using ImageJ. Using tools of 3D projection and TransformJ in the software, the dendritic area in 2 dimensions was defined and measured.

## Visual stimuli

Visual stimuli were generated using the Matlab and psycho-toolbox (version 3.0.16). This script elicited approximately 5-volt pluses through a line printer terminal (LPT) from the computer to synchronize the visual stimuli with Fluoview 1000. For the OKR behavioral test, larvae were adapted to the new environment for about 5 minutes before recording. The shift grating, at maximum contrast (1.0) and optimized spatial resolutions (0.026 cycle/degree), was projected to the screen (E Color 216 Full White Diffusion Gel Sheet, Rosco) on the cylinder. Grating was projected to the screen on the cylinder. With binocular full field stimuli (speed: 16.3 cm/s), the visual distance to cylinder was 1.5 cm. For *in vivo* two-photon calcium imaging,

moving bars (bar width: 50 pixels, visual angle: 50˚ given the visual distance of 0.36 cm, velocity: 1.63 cm/s, full-field screen illumination) of 12 directions (0˚: leftward motion across all samples; ones between 0 and 360 degrees at an interval of 30 degrees; presented in a random order) were generated in every single trials, and each experiment included 4 trials. For object size preferential test, moving bars in the small range (bar width: 2, 5, and 10 pixels, visual angle: 1˚/pixel given visual distance of 0.36 cm) and the large range (bar width: 10, 50, and 250 pixels, visual angle: 1˚/pixel given visual distance of 0.36 cm) in a full-field screen were applied. Four trials of each bar width at each direction were given as repetitive stimulus. An adaption time (time window for calcium response decaying) of 20 seconds was set between 2 visual stimuli epoch. For ON and OFF light response, a full-field light spot was projected to the screen for 3 seconds. An adaption time of 20 seconds was set between 2 light flashes with black background.

## *In vivo* two-photon calcium imaging

Calcium activity recording was performed with a 25×/1.05 NA water-immersion objective using an Olympus Fluoview 1000 two-photon microscope (Tokyo, Japan). Excitation was provided by a Chameleon laser (Coherent) tuned to 930 nm. Time-series of visually evoked calcium responses in *bhlhe22*+ and *sox2*+ ACs were acquired at a rate of approximately 4.0 to 4.8 Hz and 0.636 × 0.636 μm/pixel resolution (image size: 96 × 92 pixels). Plasmid 14uas: sypb-Gcamp6s was injected to TgBAC(*bhlhe22*: *gal4ff*) and TgBAC(*sox2*: *gal4ff*) to obtain sparsely labeling of single cell. Larvae were brought up in 0.003% phenylthiourea from 8 hpf to 5 to 8 dpf before imaging. They were first paralyzed by α-bungarotoxin (Tocris, 1 mg/mL) and mounted dorsal up by 2.5% low melting agarose. Larvae were placed in an imaging dish and immersed with embryo medium after the agarose was solidified. Imaging dish was placed on a 7.0×12.4 cm liquid crystal display (LCD) screen projected with moving bar. The screen was covered by red diffusive filters (manufacturer: shunyuan, wavelength: approximately 600 to 780 nm, transmission efficiency: 7.01%, black background:0.017 cd/m$^2$, visual stimuli: 10.567 cd/m$^2$) to alleviate the background light from screen. Vertical distance from larvae lens to the screen was 0.5 mm (visual angle: 178˚ × 179˚). To ensure the retina to adapt the black background, laser scanning of 60 seconds was inserted before the initiation of visual stimuli.

## Functional analysis

The image processing of calcium responses was conducted as previously described [43]. Iterative Grubbs' test (alpha = 0.01, GraphPad Prism) was applied to remove the outliers of data in statistics. Images were processed with ImageJ. Three ROIs of blank area were selected for background subtraction. For calcium responses detected by 14uas: sypb-gcamp6s plasmid, segmented all local dendrites with tiling ROIs (0.19 × 0.19 μm) in the image from each cell of *bhlhe22*+ and *sox2*+ type for further analysis. The F value was calculated by grayscale mean values of each ROI. Approximately 3.5 seconds of adaption period (20 seconds) before visual stimuli was selected as baseline epoch (averaged grayscale value: F0) of each ROI, and dF/F0 (dF = F–F0) was then calculated at each time point. A visual response was identified when it meets following 3 conditions: (1) the dF in the epoch of stimuli was greater than 5×SD of the baseline; (2) the occurrence should be more than 2 times out of 4 repeated trials; and (3) according to the previous study in the zebrafish retina [43], outliers of trials were removed by threshold of dF = 5×SDs in our study, and variance of dF in visual stimuli epoch of trials exceed 4×SDs was removed as outliers. Average value of trials after removing outliers was calculated, and it was the final raw response of each ROI. Peak amplitude during visual stimuli of final raw response was set as the representative response (dF/F0) of the ROI. The raw response was normalized to $\sum \left(\frac{dF}{F0}\right)_i$, where *i* is 1 to 12, corresponding to 12 directions spanning 0˚ to

330˚ at 30˚ interval. r is the normalized response, and the preferred direction was the direction of vector sum: $\sum \rightarrow r_i$ [77,78]. DSI was calculated as follows:

$$DSI = \frac{Pref - Null}{Pref + Null}$$

where Pref is the larger raw response dF/F0 in the stimulus direction close to the preferred direction, and null is the response in the opposite stimulus direction. Direction selectivity threshold is DSI > = 0.5 [43].

For calcium responses detected by Tg(*uas*: *Gcamp6s*), individual soma of *bhlhe22*+ and *sox2*+ ACs was selected as a ROI for further analysis. The last 5 seconds of the adaption period (20 seconds) in each epoch was set as the baseline (averaged grayscale value: F0) of each ROI, and dF/(F0+offset) (dF = F − F0) was then calculated at each time point. Maximum value of averaged amplitude of trials in the epoch of visual stimuli was set as the response amplitude.

### Peak time and rising time

Responsive trials in all bar sizes were pooled to evaluate the peak time and rising time of the 2 types of cells. These epoch data were smoothed by gaussian method (in window 25) using Matlab. With the smoothed dataset, 20% dF/F0 of maximum amplitude time point was set as the timing when the stimuli entered the receptive field of cells as previously described [53]. Nevertheless, if this time point was not in the visual stimuli, these trials were excluded from the pool. Peak time point was the time point when the response reached to maximum amplitude in the stimuli epoch, and the rising time was the duration from 20% dF/F0 of maximum amplitude to 100%.

### *bhlhe22*+ ACs and *sox2*+ ACs ablation

For *bhlhe22*+ ACs ablation, 8 sgRNAs of *bhlhe22* and *bhlhe23* were mixed and coinjected with Cas9 protein into embryos of TgBAC(*bhlhe23*: *mNeonGreen*) at one-cell stage. The control group of TgBAC(*bhlhe23*: *mNeonGreen*) was coinjected with control sgRNAs and Cas9 protein. *bhlhe22*+ ACs thoroughly removed ones were selected for the behavior test. Selected larvae zebrafish were continuing reared in the incubator until approximately 7 to 8 dpf before behavioral recording. Nitroreductases (nfsB)–metronidazole (MTZ) system was applied to specifically ablate *sox2*+ ACs. Cells expressing nfsB protein were killed with MTZ treatment [79]. MTZ (Sigma-Aldrich) was freshly prepared and stored at 20 mM in the embryo medium in the dark. Firstly, TgBAC(*sox2*: *gal4ff,uas:nfsB-mCherry*) zebrafish were screened according to the mCherry expression patterns in the retina at 4 dpf. The most densely labelled ones were selected for further MTZ treatment. MTZ was then immediately added into the embryo medium at the working concentration of 5 mM. Meanwhile, the control group was not added with MTZ, and zebrafish of TgBAC(*sox2*: *gal4ff,uas:nfsB-mCherry*) were simply cultured in the embryo medium. From 4 to 6 dpf, they were treated with the drug, followed by being reared in the incubator with the normal light/dark cycles. MTZ was refreshed twice a day. At 6 dpf, the drug was washed out and replaced with the fresh embryo medium, and both groups were recovered for 24 hours before behavioral tests. For quantification, a tangential two-dimensional area of 30 × 100 μm was selected from a frame of whole mount and frozen section of Tg (*ptf1a*: *EGFP*) and TgBAC(*sox2:gal4ff,uas:nfsB-mCherry*) retina, respectively.

### Optokinetic reflex recording apparatus setup

The imaging chamber was a specifically designed cylinder, which upheld larvae in the middle part. Visual stimuli were generated using the matlab template script. This script was able to

elicit approximately 5 V through LPT from computer to camera (Cool*SNAP* KINO, photometrics) to give the external trigger for recording with the micromanager software (ImageJ). By this way, when visual stimuli were generated, the camera connected to the computer was triggered to initiate the recording synchronously. The sampling of the OKR recording was 30 frames/second.

### Optokinetic reflex recording sample preparation

Larvae zebrafish for the OKR behavioral test were cultured in standard protocol. They were embedded in 0.8% agarose without anesthesia. After the agarose was solidified, agarose around rostrum and eyes was removed immediately, but cautiously leaving the other part of body embedded. Agarose with larvae were then immersed in the embryo medium on a glass slide of the imaging chamber.

### Optokinetic reflex recording analysis

When the video was generated with images recorded by ImageJ (version: 1.48v), visual angles of eyes moving with shift grating were tracked using OKR track Matlab codes [80]. Based on the analysis method in this study, there are 3 steps for analyzing the OKR performance. Step 1: Plot the trajectory and label peaks of the saccade. Step 2: Count the saccade number manually using tables generated in step 1. Step 3: Summarize all the data and do statistic. N: saccade number, T: 40 seconds (there are 5 seconds of blank control before and after grating). $D_t$: visual degree in the rough of a saccade, $D_p$: visual degree in the peak of the saccade.

$$\text{Saccade frequency} = N/T$$

$$\text{Saccade amplitude} = |D_t - D_p|$$

The mean of all saccade amplitude of 2 eyes was calculated as the individual saccade amplitude of a sample.

### Image analysis and quantification

For live samples, the cells in a volume of $80 \times 90 \times 10$ μm$^3$ (height × width × z stack) region were quantified. For fixed samples, the cells in an area of $100 \times 100$ μm$^2$ (height × width) were quantified. Mann–Whitney test was used, and data were presented as mean ± SD. GraphPad 7.0 and Matlab R2018b are used to analyze the data.

The total numbers of cells counted for each animal used for the statistical quantification have been summarized in S3 Data.

## Supporting information

**S1 Fig. Single-cell transcriptome analysis identifies 2 GABAergic/cholinergic AC types.** (**A**) t-SNE plot showing 19 clusters of the zebrafish retina (72 hpf). Each cluster is identified as one cell type according to highly expressed cluster-specific marker genes. (**B**) t-SNE feature plot showing *elval3* and *tfap2a* expression level in 19 clusters of the zebrafish retina. (**C**) t-SNE feature plot showing *bhlhe22* and *bhlhe23* expression level. (**D**) t-SNE feature plot showing *sox2* and *isl1a* expression level. (**E**) t-SNE feature plot showing *fezf1* and *tenm3* expression level. (**F**) The validation of riboprobe sequences for marker TFs of cluster AC4 (*bhlhe22* and *bhlhe23*) and AC5 (*sox2* and *isl1a*). (**G**) Expression patterns of riboprobes in (**F**) using *in situ* hybridization (magenta) combined with ChAT immunostaining (green) of the 5-dpf zebrafish. (**H**) Validation of cluster AC5 marker *isl1a* (magenta) using *in situ* hybridization combined

with SOX2 (green) immunostaining. (**I**) Cell fraction composition analysis of *sox2*+ cells in (**H**). Solid white arrow head indicated *sox2*+ *isl1a*+ cells. The data underlying this figure can be found in S3 Data. Scale bars, 10 μm. AC, amacrine cell; ChAT, choline acetyltransferase; dAC, displaced amacrine cell; dpf, days post-fertilization; GCL, ganglion cell layer; IPL, inner plexiform layer; TF, transcription factor.
(PDF)

**S2 Fig. Efficiency verification of gene disruption and fate specification between 2 GABAergic/cholinergic AC types.** (**A**) Statistics of *bhlhe22*, *bhlhe23*, and *sox2* disruption efficiency and gene disruption pattern through 4 CRISPR/Cas9 ribonucleoprotein complexes injection in founder embryos. (**B**) Examples of 10 alleles around 4 targeted sites of ORF of *bhlhe22*, *bhlhe23*, and *sox2*. First line is the WT sequence. Blue rectangle, sgRNA-targeted sites; red rectangle, mismatches. (**C**) Representative images showing ChAT immunostaining pattern of TgBAC(*sox2*: *mNeonGreen,bhlhe23:mRuby3*) of wild-type (up) and *sox2* disrupted (middle) larval fish, and representative images showing SOX2 and ChAT immunostaining pattern of TgBAC(*bhlhe23*: *mNeonGreen*) after *bhlhe22* and *bhlhe23* codisrupted (bottom). Hollow white arrow head indicated ChAT+ *sox2*+ cells and solid white arrow head indicated ChAT+ *bhlhe22*+ cells. (**D**) Bar plot showing cell density of 3 types of cholinergic cells (ChAT+ cells) in (**C**). (**E**) Bar plot showing cholinergic cell composition of 3 groups in (**C**). The data underlying this figure can be found in S3 Data. Data are presented as mean ± SD, Mann–Whitney test. *** $p < 0.001$, **** $p < 0.0001$. Scale bars, 10 μm. AC, amacrine cell; ChAT, choline acetyltransferase; ORF, open reading frame; sgRNA, small guide RNA; WT, wild type.
(PDF)

**S3 Fig. Fate specification to other retinal cell types of 2 GABAergic/cholinergic AC types after gene disruption.** (**A**) Representative images showing patterns of *sox2*+ ACs, *sox2*+ MCs and GS+ MCs after disruption of *sox2* in TgBAC(*glyt1*: *EGFP*). (**B**) Quantification in (**A**). (**C**) Representative images showing Sofa1 fish (generated by crossing Atoh7: gapRFP, Ptf1a: cytGFP, and Crx:gapCFP) patterns after disruption of *sox2* and *bhlhe22/bhlhe23*. (**D**) Cell fraction composition of major retinal cell types of 3 groups in (**C**). The data underlying this figure can be found in S3 Data. Data are presented as mean ± SD, Mann–Whitney test. * $0.01 < = p < 0.05$, **** $p < 0.0001$. Scale bars, 10 μm. AC, amacrine cell; BC, bipolar cell; dAC, displaced amacrine cell; GCL, ganglion cell layer; HC, horizontal cell; IPL, inner plexiform layer; MC, müller cell; RGC, retinal ganglion cell; PR, photoreceptor cell; sgRNA, small guide RNA.
(PDF)

**S4 Fig. Morphology of individual *bhlhe22*+ and *sox2*+ AC.** (**A**) Schematic design (up) of TgBAC(*bhlhe23*: *gal4,uas:kaede*) and the representative image (down). (**B**) Schematic design (up) of TgBAC(*sox2*: *gal4ff,uas:kaede*) and the representative images (down). (**C**) Representative images showing the ON and OFF subtype dendritic morphology of *bhlhe22*+ and *sox2*+ ACs. Images are captured from larval fish of 4 to 5 dpf. Scale bars, 10 μm.
(PDF)

**S5 Fig. Response patterns to light ON and OFF.** (**A**) Heatmap showing the response of all DS *bhlhe22*+ ROIs in (Fig 5C). (**B**) Heatmap showing the response of all DS *sox2*+ ROIs in (Fig 5G). (**C**) Violin plot showing peak amplitude of *sox2*+ and *bhlhe22*+ responsive ROIs to moving bar. The gray lines indicate the quartiles, and the red lines indicate the median. Mann–Whitney test. **** $p < 0.0001$. (**D**) Violin plot showing response latency of light on response of *sox2*+ and *bhlhe22*+ responsive trials. Mann–Whitney test. (**E**) Polar plot of distribution of DSI and PD of *bhlhe22*+ and *sox2*+ DS ROIs. Scale bar, DSI value = 0.5. (**F**) Bar plot of responses of DS *bhlhe22*+ and *sox2*+ ROIs in Fig 5C and 5G. Data are shown in mean ± SD.

Mann–Whitney test. (**G**) Violin plot showing DSI of DS and non-DS ROIs in **Fig 5C** and **5G**. Mann–Whitney test. (**H**) Histogram polar plot of PD distribution of DS ROIs of 2 types of cells. Scale bar, ROI number = 50. Cumulative frequency comparison with Kolmogorov–Smirnov test. (**I and J**) Heatmap of response to light on (**I**, ON Res.) and off (**J**, OFF Res.) of *bhlhe22*+ ROIs. The schematic showing visual stimuli paradigm of full-field screen spot was on top of heatmap. (**K and L**) Heatmap of response to light on (**K**, ON Res.) and off (**L**, OFF Res.) of *sox2*+ ROIs. (**M**) Statistics of response patterns in (**I** to **L**). The data underlying this figure can be found in S3 Data. DS, direction selective; DSI, direction selectivity index; PD, preferential direction; ROI, region of interest.
(PDF)

**S6 Fig. Cell numbers of other cell types in the retina after the ablation of either type of GABAergic/cholinergic ACs.** (**A**) Workflow(upper) and representative images (bottom) showing the OKR assay of *bhlhe22*+ ACs-ablated zebrafish. (**B**) Workflow (upper) and representative images (bottom) showing the OKR assay of *sox2*+ AC-ablated zebrafish. (**C and D**) Representative image (**C**) and quantification (**D**) of a section (30 × 100 μm) of whole-mount Tg(*ptf1a*: *EGFP*) retina with *bhlhe22*/*bhlhe23* codisruption to ablate *bhlhe22*+ ACs. Cell number and types were defined by cell body location in layers. (**E and F**) Representative image (**E**) and quantification (**F**) a frozen section of (30 × 100 μm) TgBAC(*sox2*: *gal4ff*, *uas:nfsB-mCherry*) retina with MTZ to ablate *sox2*+ ACs. Cell number and types were defined by cell body location in layers. (**G and H**) Representative image labeling with MCs marker anti-GS (**G**) and quantification (**H**) of a frozen section of (100 × 100 μm) wild-type with *bhlhe22*/*bhlhe23* codisruption to ablate *bhlhe22*+ ACs. (**I and J**) Representative image (**I**) and quantification (**J**) of a frozen section of (100 × 100 μm) TgBAC(*sox2:gal4ff,uas:nfsB-mCherry*) labeling with GS after MTZ treatment. The data underlying this figure can be found in S3 Data. Data are collected from 5 to 8 dpf the larval zebrafish, presented as mean ± SD, Mann–Whitney test. **** $p < 0.0001$. Scale bars, 10 μm. AC, amacrine cell; BP, bipolar cell; GCL, ganglion cell layer; HC, horizontal cell; INL, inner nuclear layer; IPL, inner plexiform layer; MC, müller cell; MTZ, metronidazole; OKR, optokinetic reflex; PR, photoreceptor cell; RGC, retinal ganglion cell; sgRNA, small guide RNA.
(PDF)

**S7 Fig. Response patterns of responsive *bhlhe22*+ and *sox2*+ type cells.** (**A and B**) Heatmap showing the response pattern of all responsive *bhlhe22*+(**A**) and *sox2*+(**B**) ACs in **Fig 6E** and **6F**. (**C**) Schematic of a smoothed responsive trial for peak time and rising time. (**D**) Bar plot showing peak time of all smoothed responsive trials of 2 types of ACs as described in (**C**). (**E**) Bar plot showing rising time of all smoothed responsive trials of 2 types of ACs as described in (**C**). The data underlying this figure can be found in S3 Data. Data are presented as mean and SD. Mann–Whitney test. **** $p < 0.0001$.
(PDF)

**S8 Fig. Schematic of GABAergic/cholinergic ACs.** (**A and B**) Schematic showing cell-body positioning of zebrafish GABAergic/cholinergic ACs (**A**) and mouse SACs (**B**). (**C and D**) Schematic showing dendritic arborization in zebrafish GABAergic/cholinergic ACs (**C**) and mouse SACs (**D**).
(PDF)

**S1 Data. Top 100 marker genes of 19 clusters in S1 Fig.**
(XLSX)

**S2 Data. Primers of plasmid, BAC constructs, sgRNA, and probes.**
(XLSX)

**S3 Data. Statistical details of individual sample.**
(XLSX)

**S1 Movie. Z stack of transgenic line TgBAC(*bhlhe23*: *mNeonGreen*).** Z stack animation from ventral to dorsal of brain and retina of 5-dpf larval fish. A, anterior; P, posterior. Scale bars, 50 μm.
(AVI)

**S2 Movie. Z stack of transgenic line TgBAC(*bhlhe22*: *mNeonGreen*).** Z stack animation from ventral to dorsal of brain and retina of 5-dpf larval fish. A, anterior; P, posterior. Scale bars, 50 μm.
(AVI)

**S3 Movie. Z stack of transgenic line TgBAC(*sox2*: *mNeonGreen*).** Z stack animation from ventral to dorsal of brain and retina of 5-dpf larval fish. A, anterior, P, posterior. Scale bars, 50 μm.
(AVI)

**S4 Movie. Optokinetic reflex of *bhlhe22*+ type-ablated TgBAC(*bhlhe23*: *mNeonGreen*) larva.** OKR performance of 7-dpf TgBAC (*bhlhe23*: *mNeonGreen*) larva with *bhlhe22*/*bhlhe23* codisruption and stimulated using paradigm showing in **Fig 5M**. Scale bars, 100 μm.
(AVI)

**S5 Movie. Optokinetic reflex of control sgRNA treated TgBAC(*bhlhe23*: *mNeonGreen*) larva.** OKR performance of 8-dpf TgBAC (*bhlhe23*: *mNeonGreen*) larva with scrambled sgRNAs (control sgRNAs) treatment and stimulated using paradigm showing in **Fig 5M**. Scale bars, 100 μm.
(AVI)

**S6 Movie. Optokinetic reflex of *sox2*+ type-ablated TgBAC(*sox2*: *gal4ff*;*uas*:*nfsB-mNeon-Green*) larva.** OKR performance of 7-dpf TgBAC (*sox2*: *gal4ff*; *uas*:*nfsB-mNeonGreen*) larva treated with MTZ and stimulated using paradigm showing in **Fig 5M**. Scale bars, 100 μm.
(AVI)

**S7 Movie. Optokinetic reflex of TgBAC(*sox2*: *gal4ff*;*uas*:*nfsB-mNeonGreen*) larva without MTZ treatment.** OKR performance of 7-dpf TgBAC (*sox2*: *gal4ff*; *uas*:*nfsB-mNeonGreen*) larva without MTZ and with stimulation showing in **Fig 5M**. Scale bars, 100 μm.
(AVI)

## Acknowledgments

We are grateful to J. Chu, Y. Mu, H. S. Yao, Y. F. Zhang, and J. L. Du for reagents and suggestions, and H. Gao, L. Zhang, and H. Qian for data analysis. We also thank He lab members for experimental assistance.

## Author Contributions

**Conceptualization:** Yan Li, Jie He.

**Data curation:** Yan Li.

**Formal analysis:** Yan Li.

**Funding acquisition:** Jie He.

**Investigation:** Yan Li.

**Methodology:** Yan Li, Xinling Jia, Xiaoying Qiu.

**Project administration:** Yan Li, Jie He.

**Resources:** Shuguang Yu, Jie He.

**Software:** Yan Li.

**Supervision:** Jie He.

**Validation:** Yan Li.

**Visualization:** Yan Li, Shuguang Yu.

**Writing – original draft:** Yan Li.

**Writing – review & editing:** Yan Li, Jie He.

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
