## [Editor Report · Decision Letter 0]

11 May 2023

Dear Dr He, 

Thank you for submitting your manuscript entitled "Defining morphologically and genetically distinct direction-selective amacrine cells in the vertebrate retina" for consideration as a Research Article by PLOS Biology.

Your manuscript has now been evaluated by the PLOS Biology editorial staff as well as by an academic editor with relevant expertise and I am writing to let you know that we would like to send your submission out for external peer review.

Once your full submission is complete, your paper will undergo a series of checks in preparation for peer review. After your manuscript has passed the checks it will be sent out for review. To provide the metadata for your submission, please Login to Editorial Manager (https://www.editorialmanager.com/pbiology) within two working days, i.e. by May 15 2023 11:59PM.

Kind regards,

Lucas

Lucas Smith, Ph.D.

Associate Editor

PLOS Biology

lsmith@plos.org

---

## [Decision Letter · Decision Letter 1]

13 Jul 2023

Dear Dr He,

Thank you for your patience while your manuscript "Defining morphologically and genetically distinct direction-selective amacrine cells in the vertebrate retina" was peer-reviewed at PLOS Biology. It has now been evaluated by the PLOS Biology editors, an Academic Editor with relevant expertise, and by several independent reviewers. As you will see in their comments below, the reviewers find the study interesting and generally well done - however they have a number of comments and suggestions that we think should be addressed in order to strengthen the study. In light of the reviews, we would like to invite you to revise the work to thoroughly address the reviewers' reports. 

Given the extent of revision needed, we cannot make a decision about publication until we have seen the revised manuscript and your response to the reviewers' comments. Your revised manuscript is likely to be sent for further evaluation by all or a subset of the reviewers.

**IMPORTANT - SUBMITTING YOUR REVISION**

*Re-submission Checklist*

*Published Peer Review*

*PLOS Data Policy*

*Blot and Gel Data Policy*

Sincerely,

Luke

Lucas Smith, Ph.D.

Senior Editor

PLOS Biology

lsmith@plos.org

REVIEWS:

Reviewer #1, Juan Angueyra (note, Reviewer 1 has signed this review): 

SUMMARY. 

Li et al. describe two genetically and morphologically distinct amacrine-cell (or AC) types to have a role in direction selectivity in the zebrafish retina. Like the mouse starburst amacrine cell (or SAC), the sox2+ AC seems to be a symmetrical, narrow-field AC that is GABAergic and cholinergic. Interestingly, the authors identified a novel direction-selective AC that is also GABAergic and cholinergic and that shows direction-selectivity; this novel AC is characterized by expression of the transcription factor bhlhe22+. Both cell types can respond to moving bars of various sizes; the bhlhe22+ cells seem to prefer larger moving stimuli in comparison to the sox2+ cells. The authors show convincing differences between these two cell types by transcriptomics, genetics, morphology and function, providing a well-rounded starting point for future characterization of cholinergic cells and direction-selectivity in the zebrafish retina. The experiments are thorough, well-controlled and analyzed. Collectively, the authors provide an insightful and significant piece to the expanding understanding of retinal-cell type diversity in vertebrates. The optokinetic and optomotor responses have become standard assays to evaluate visual function and perception across vertebrate animal models; understanding how direction selectivity arises in the retina is crucial to inform our understanding of the central processing of visual information.

SUGGESTIONS.

We consider that the results presented here very convincingly delineate the differences between these novel subtypes of cholinergic cells in the zebrafish retina. As the paper stands, the transcriptomic, genetic, morphologic data and the mutations in the respective transcription factors clearly support the main claims of this paper. These results are important for out field.

Most of our concerns lie in the characterization of the direction selectivity. We explain these below.

Selection of responsive cells

- It is difficult to understand if the population data presented in Figure 4 derives from single ROIs, or if they represent averages across the 3 ROIs for a single cell. Methods mention "To evaluate the variance of preferential directions of various local dendritic regions from each cell，the standard deviation of tuned direction of cells with responsive ROIs more than 2 was analyzed" but these data are not presented or mentioned in results. Given the mammalian work on SACs, it would be presumed that individual dendrites may work independently and have disparate preferred directions, so that ROI data should not be pooled. Please make clarifications and consider changing labels from cell to ROI where appropriate

Selection of ROIs:

- What was the random selection process for ROIs?

- Is 0° standardized between larvae? is it rightward motion? please explain

- Text describes that the response in sox2+ cells is smaller than for bhlhe22+ cells but example panels B and D in Figure 4 show the opposite

Possible direct stimulation of cells:

- How was the size of the bars determined and how does the size of the bar match the receptive field, or the dendritic arbor, of both cell types?

- In our experience, while recording calcium-responses in the retina in a PTU-treated fish, it is possible (and likely) that stimulation of photoreceptor occurs directly with light that traverses larval tissues and enters the back of the eye (which has been made transparent) and is absorbed by outer segments. Most of the optic tectum work is perhaps protected from this issue because they tend to keep eye pigmentation. We have recorded robust retinal responses in larvae positioned on their side and with stimulation through the condenser, which precludes light stimulating the retina by entering through the lens.

- Given the size of the dendritic fields of these cells (less than a 100 microns), and the speed of the bars (16300 microns per second) and screen refresh rate (60 Hz?), the bar edge moves ~270 microns between frames, potentially covering all cones that provide input to these cells in a single frame (if they were stimulated directly and not by light entering through the lens)

- Can the authors make any claims about the location of each cell's receptive field with the available data? Can these be estimated?

- Alternatively, can the authors test if direct stimulation of photoreceptors is an issue by flashing small spots underneath the larvae? Or, can the stimulus be blanked underneath the larvae while direction selective responses are being measured?

Size selectivity

- For size selectivity and Figure 6, the analysis of DSI seem to switch from individual ROIs to cell bodies. Is this correct? Can the authors explain the rationale behind this decision?

- In panels I and K, the calculated DSI are above 0.20 for the example cells. For what bar width are these example responses?

- The average DSI across all bar widths is well below 0.20 (Panels M and N). Are the example traces from far outliers in the dataset?

- If DSI threshold in Figure 5 was 0.1, should the interpretation of these data be that there is little to no direction-selective responses to these bars as measured in this figure?

Statistical analysis: - Mann-Whitney tests are appropriate for single pair-wise comparisons but are likely to give false positives when repeatedly used for multiple pairwise in groups larger than 2. - Consider using a Kruskal Wallis test and a subsequent post-hoc test where appropriate.

Line 88: " At 72 hpf, retina development is thought to be mostly completed in zebrafish [38]" This statement should be modified. At 72 dpf, most retinal cells are still quite immature; a prime example are photoreceptors which have short outer segments, poor light responses correlating to low expression of phototransduction genes and with a mosaic that is very disorganized.

Line 178 - 186: It is not made clear for readers that the experiments for "The generation of two AC types requires distinct sets of TFs" section derive from F0/G0 mutants, which are genetic mosaics, and not germ line mutants.

The variability is dendritic field size of bhlhe22+ cells is quite large. The smaller cells in this "wide-field" class rival the size of the sox2+ "narrow-field" cells. These all correspond to 5 dpf larvae, correct? Is there a systematic difference between the ON and the OFF subtypes? Is this same range seen in cells that responses were recorded from? Do responses match dendritic size characteristics at all?

The authors suggest the evidence illustrates sox2+ cells in zebrafish to be the evolutionary counterpart to the mouse SAC cells. Authors may consider assessing the coverage factor and tiling of the sox2+ cells in the zebrafish. Are the sox2+ cells as densely tiled as the SACs in mice? What about the bhlhe22+ cells?

Figure 2, panel F: there is not enough information to decode this panel in results or legend. The description for panel F is duplicated and not very informative. Why are there 2 bars? what is 100% for each one? why do they share an n?

Figure 2, panel K: remove yellow dotted outline in Gad65/67 panel 

Figure S1, panel H: what does "dAC" stand for? 

Figure 4 and Figure S4, panel A/B: kaede is mispelled 

Figure 4, panel E: "Placed" is an awkward term for this. Perhaps: "Cell body in INL" vs. "Cell body in GCL (displaced)"?

Figure 4, panel G: the data suggests that there is a loss of cells on average and not just a lack of displaced cells. Are the differences in the sum of ON and OFF sox2+ cells significant?

Methods:

Line 59: How long were samples fixed for?

Line 66 - 68: provide details about antibodies including species, catalog and lot number.

Missing details for OKR: how far was the cylinder from the larvae? how fast did the grating move? was it reversed? for how long? can you provide details about the screen (make, spectral compositions, etc.)?

Missing details for Visual stimulation under 2P: How far was the larvae from the screen surface? how many degrees of visual angle are covered by the screen?

In general, the introduction and methods could use style editing and grammar corrections that would help with clarity. A none comprehensive list is below.

MINOR EDITS.

Line 37 - change to "… (SACs), which are also cholinergic [3-8]." Line 42 - 44 - change to "in the mouse retina, expression of Sox2 and Fezf1 were required for appropriate cell body positioning of ON SACs [12,13]."

Line 44 - add coma "showed that OFF, but not the ON subtype,…"

Line 45- 46 - Run on sentence. Modify to "…from non-SACs. These inputs, involved in centrifugal direction selectivity, suggest different…"

Line 49 - Change "were" to "are" Line 58 - Unclear wording. Change to "…[22-24]; which are a diverse cell type [25,26]."

Line 60 - Add semicolon. "…neurite arborization; including: narrow-field…"

Line 62 - delete "either" Line 62 - Run on sentence. Change to "…[29]; however, there are a few amacrine cells that…"

Line 65 - Organization. Structure to " …arbors with some overlapping and are born earlier…"

Line 66 - "Cholinergic ACs are a subpopulation of GABAergic ACs" 

Line 70 - 71 - Modify wording to "Additionally, we performed…"

Line 74 - change "further" to "furthermore" Line 80 - wording unclear. Change "showed higher responses" to "responded more robustly to…" 

Line 81 - Change "the broader size range…" to "a broader size range" 

Line 89 to 91 - Change to: "From the 19 retinal clusters we identified 5 AC clusters based on co-expression of elval3 and tfap2a; these 5 AC clusters could be readily differentiated by the expression of marker genes (...)" 

Line 95 - Noun is singular. Change "were" to "was"

Line 96 - define "MC" here.

Line 98 - Use ChAT. Already defined acronym in line 52 Line 100 - delete "were" in the area: "type were resided in both…"

Line 125 - Consider changing "mark" to "identify"

Line 126 - delete "either"

Line 126 - mNeonGreen is misspelled Line 128 - rephrase for clarity. "…specifically marked a subset of ChAT+ and Gad65/67+ ACs located in the INL…"

Line 129 - add "the" - "together with the in situ results…"

Line 130 - rephrase for clarity. "…specifically marked the bhlhe22+…"

Line 131 - rephrase for clarity.

Line 132 - add semicolon. "…bipolar cells(Fig 2E and 2F); which is consistent…"

Line 133 - rephrase for clarity. "specifically marked a ChAT+ and Gad65/67+ subset of ACs located in the INL…; which is consistent with…"

Line 135 - reword for clarity. "…were co-labeled with a SOX2 antibody…"

Line 137 - Change "besides" to "additionally"

Line 139 - Reword for clarity and add "the" before the word "two". "Furthermore, crossing TgBAC… show that the two AC types…"

Line 211- Unclear sentence. Reword for clarity.

Line 212 - Change 'it' to 'this' 

Line 212 - Change wording. "…we examined the influence of disrupting bhlhe22/bhlhe23 or sox2…"

Line 217 - Reword for clarity. "we could not…that SoFal lacked the sensitivity to detect changes within retinal subtypes."

Line 223 - add "lines" - "using newly-generated lines…"

Line 225 - add "the" - "…, the dendrites…"

Line 226 - reword for simplification. "… starburst-like pattern reminiscent of mouse SACs [16]." Line 316 - use 'OKR' abbreviation already defined in line 75.

Line 316 - change "is the behavior" to "is a behavior" Line 390 - change 'it' to 'this' Line 459 - change 'the' to 'a' Line 468 - Reword for clarity. Change 'higher' to 'more robust'

Line 473 - Simplify wording. "which express sox2. [43]."

Line 474 - add colon. - "… including: cat…"

Line 475 - Simplify for clarity. Start sentence at "Although cholinergic ACs are observed in many non-mammalian species, such as: chick…"

Line 479 - Consider using different wording for 'placed' and 'displaced' cells or clearly defining them in this context.

Line 480 - Consider rewording for clarity. "led to inappropriate soma placement in the INL" ?

Line 499 - delete 'to' - "presynaptic bipolar cells (BPs)…"

Line 503 - add 's' to AC Line 505 - delete 'and' - "…, thereby resulting in…"

Line 506 - what does 'more-fined" mean here? Consider rewording.

Line 507 - add 's' to population

Line 508 - Reword for clarity and simplification. "…subtypes also lack specific ON or OFF responses."

Line 509 - Run-on sentence. End at "response." - "This requires future efforts…"

Line 509 - change 'in' to 'of' - "difficulty of genetically marking…"

Line 514 - change 'on' to 'of' - "the advantage of…"

Line 517 - delete 'the' - "… generated larger responses…"

Line 517 - delete 'of' - "…same size objects…"

Line 518 - add 'the' - "… the larger dendrites…"

Line 518 - change 'titling' to 'tiling'

Line 520 - add 's' to type Line 521 - delete 'either'

Line 521 - Change wording for clarity. Change 'more' to 'additional' or 'more numerous'

Line 522 - change 'side' to 'size'

Line 522 - reword for clarity. " …could enhance the response of the bhlhe22+ type as a function of bar size."

Line 526 - Change 'may better receive' to 'may receive more'

Line 526 - delete "dendrites, which could result" change to 'dendrites resulting in an…"

Reviewer #2, Takeshi Yoshimatsu (note, Reviewer 2 has signed this review): This manuscript by Li Y, et al. reports the discovery of two types of direction-selective (DS) amacrine cells in larval zebrafish. The genetic approaches to identify DS amacrine cells are very thorough and the amount of work done is impressive. In particular, the finding that one of the DS types is similar to the mammalian DS amacrine cells (aka starburst amacrine cells) in terms of the molecular marker and the developmental origin (fexf2 dependent cell locarization) is interesting. The authors further characterize the morphology and the properties of direction selectivity in these DS amacriine cells and found that one type has smaller dendritic field size and responds robustly to smaller moving objects, whereas the other type has larger dendritic field size, larger responses, and tuned to larger moving bars. Finally, the authors tested the roles of these DS amacrine cells in visual behaviors using an optokinetic response assay. This revealed that that both types are required for the normal frequency of saccade eye movements, but not for the normal saccade amplitude. Taken together, this manuscript reports the conserved and a novel type of DS amacrine cells in the vertebrate retina. These findings are of interest to broad audience including retinal physiology and visual ecology. However, I find that the Methods section in the supporting information needs much improvement and some clarification is required in the results due to the lack of some information in the methods.

1. Did the authors verify the sequences of the riboprobe? According to the methods section, the DNA template for the probe synthesis was prepared by PCR reaction from cDNA. This means that the DNA template sequence was not confirmed.

2. The visual stimuli parameters are unclear. For object size preferential test, the bar widths are described as (bar width: 2,5, and 10 pixels, 0.07 cm/pixel) for the small range. However, in Fig. 6 E, there are three bar widths used. For the big range, what was the size of the pixel? Finally, the size on the screen is not very informative. The authors should provide the size in the visual angle.

3. Are the 10 pixel bar sizes in the small range and the large range the same? If so, why are the response amplitudes, responsive cell fractions, and DSI different in Fig. 6E-H, M,N?

4. In the ‘in vivo two-photon calcium imaging’ section, the resolution is written as ‘60.42×57.88 μm’ (line 121). This resolution seems too big to resolve cells…

5. The authors should clarify the location of the cells recorded in in vivo two-photon calcium imaging. Because the visual stimuli were projected beneath the fish, this information is important to know the visual angle for a given bar width. For example, if the cell was located in the dorsal region of the eye, they see the bottom of the dish where the bars with the same width would appear wider in the visual angle compared to the bars further away which is seen by the cells located close to the center of the eye.

6. This sentence is unclear ‘Time-series images process of calcium responses was performed described (132)’

7. In the ‘Functionl analysis’ section, the authors used some criteria to identify “visual response”. However, the rationale to remove ‘3),trials with peak amplitude exceed 4×SD of 140 dF/F0 of all four trials’ as outliers are unclear.

8. Line 141. ‘Peak amplitude of averaged value of all trials during visual stimuli was the raw response (dF/F0).’ This does not make sense. If this is true, there is one raw response per ROI.

9. Line 144. How the DSI is computed is unclear. Where ‘r’ is computed?

10. What is the reason to evaluate the variance of preferential directions? (Line 146)

11. Line 148, the sentence ‘For calcium response… analysis.’ Is a duplicate of the sentence in line 134.

Reviewer #3: This paper describes for the first time the presence of two amacrine cells (AC) types in the zebra fish retina that have molecular properties similar to cholinergic/GABAergic cells seen in mammalian retinas. Evidence points to a role of these cells in generating direction-selective responses within the zebra fish visual system. The circuitry generating direction-selectivity within retinal ganglion cells has been widely studied, mostly in mammalian species, as an example of complex neural processing. The paper presents some interesting new results and lays the groundwork for a comparative analysis of DS circuits in fish versus mammalian systems, which could be of general interest to neuroscientists. 

The molecular characterization of the 2 novel AC types was convincing and well supported by the data. Calcium imaging of terminal dendrites demonstrated directional responses in these ACs, but control data showing the magnitude of direction-selectivity indices (DSI's) recorded from non-directional cells under the same conditions is missing. Such control data seems essential to set the criterion DSI level for determining whether a cell is considered direction-selective or not. As it stands, the authors simply use a cut-off for the DS index of 0.1, which seems rather low. In any case, in the absence of control data, it would be helpful to know the rationale for selecting this 0.1 criterion level. 

A focus of the paper is the finding that the two AC types have rather different dendritic field sizes. This observation prompted an analysis of the size-selectivity of the two AC types. Functional differences were observed between the two AC types, however, this section was not well motivated. If the ACs are providing directional inhibition that generates directional responses in ganglion cells, it is not obvious why the cells should be tuned to the width of the stimulus bar. It would be helpful if the authors provided a stronger rationale or a proposed circuit model that could help the reader interpret the results and understand the implications of this size-sensitivity. 

A key finding, linking the activity of the two ACs to the generation of directional responses in the fish visual system is supported by the OKR experiments shown in Fig. 5J-M. These results demonstrate a rather modest but significant effect of cell ablation on the saccade frequency during OKR. I do not agree that the results support a "major" contribution, particularly regarding the sox2+ cells. In any case, the authors don't explain why one would expect a change in the frequency of saccades when directional signaling is compromised. Perhaps a more informative metric would be to measure the gain of the OKR image tracking between saccades? See Fig 1 in Yonehara et al (2018) for an example in the mouse (https://doi.org/10.1016/j.neuron.2015.11.032).

Overall, there are some very interesting new observations, but in my view the paper could be greatly improved by a substantive revision. 

Further comments and suggestions for revision:

Line 226: the resolution of the cell fills is insufficient to suggest that the sox2+ cells had a starburst-like morphology reminiscent of putative homologous cells in mouse. Individual dendritic branches are not clearly resolved, making such a judgement questionable. 

Line 227, Fig. 4B: The authors note that the dendrites of the bhlhe22+ ACs are 10 times larger than the sox2+ cells. The area of the dendritic arbor is 10-fold larger, but the dendrites are only ~2.5-fold longer. It is more conventional and useful to quote linear dimensions rather than area. This is particularly relevant to cells potentially involved in mediating direction-selectivity, which involves spatio-temporal interactions across the retina that depend on distance rather than area. 

Fig. 4E: There are no "displaced" bhlhe22+ ACs in this panel. According to the text "placed" somas are in the GCL, yet in panel C the bhlhe22+ cells are all "displaced" to the INL?

Fig. 4F: the finding is that the disruption of fezfl causes the displacement of the On-type sox2+ AC somas to the INL. I assume that the On- and Off-type sox2+ ACs were distinguished according to the placement of the somas, with On-type in the GCL and Offs in the INL? In control, the sox2+ dendritic band in Sublamina-4 (S4) is stronger than the band in S1 (Fig. 4F), consistent with the 3-fold higher cell count for On-sox2+ ACs shown in Fig. 4G. However, although the S4 band has largely disappeared in the fezfl retinas, as expected, one would expect to see a stronger band in S1, but this is not observed. Why are the dendrites of the displaced S4-ACs and S1-ACs missing from this image? Is this a representative result, and if so, is it possible that there are concomitant changes in dendritic arborization in the fezfl retinas?

Fig. 4G: In the absence of any functional data at this point, the sox2+ ACs should be labeled according to their morphological characteristic (e.g. S1 and S4 stratifying), not a presumed functional property, which may not correlate well with the anatomical properties anyhow, according to the data shown in supplemental Fig. S5.

Fig. 5B: How were response latencies measured? A 4s latency seems very long for a cell that is tasked with rapidly signaling direction and supporting OKR responses. Is this the true latency of functional responses, or is it limited somehow by the recording methods?

Fig. 5: Since the saccade frequency was significantly lower, but the saccade amplitude was unchanged in the affected animals, one might expect the eyes to reach the limit of their motion and remain pegged there during prolonged optokinetic stimulation. Was this observed?

Line 349: Please quote bar widths either in degrees of visual angle or microns on the retina.

Fig. 6I-N: The DSI's in this dataset are lower than shown earlier. Indeed, many units show DSIs that are lower than the criterion value of 0.1 used earlier. Judging from the collected data in panels M and N, the representative traces and polar plots in panels I and K look to be extreme outliers (many SD's above the mean) rather than representative. It would be more realistic to show examples that were closer to the means of the groups. 

Minor comments:

Fig. 4A: The scale bars in the two panels are not equal in length indicating slightly different magnifications. If magnification is not preserved for comparison of the cell sizes, then why not enlarge the sox2+ cell so that it can be seen better?

Check verb tense throughout the manuscript. For example, past tense is often used to refer to immutable characteristics which is not really accurate. 

Lines 69 - 83: This final paragraph seems superfluous as it simply summarizes the results.

Line 75, "Consistently, the optokinetic…" This sentence needs to be reworded.

Fig. 5C,E: although the scale bar is shown, it would be nice to know what the actual DSI's for these examples were, as is shown in Fig. 6J,L. 

Fig. 6I,K: What were the bar widths used in these examples?

---

## [Decision Letter · Decision Letter 2]

5 Dec 2023

Dear Dr He,

Thank you for your patience while we considered your revised manuscript "Defining morphologically and genetically distinct direction-selective amacrine cells in the vertebrate retina" for consideration as a Research Article at PLOS Biology. Your revised study has now been evaluated by the PLOS Biology editors, the Academic Editor and by two of the original reviewers.

As you will see in their comments, below, the reviewers are largely satisfied by the changes made in this revision. However each has noted lingering concerns that will need to be addressed before publication. Of note, Reviewer 3 has commented that the conclusions about direction selectivity in these amacrine cells is not supported by the data. S/he suggests that the findings are still interested without this conclusion, but the title, abstract, and manuscript needs to be adjusted accordingly. We have now had a chance to discuss the reviewer comments, and the additional underlying data that you provided with our Academic Editor - and after this discussion, we agree with Reviewer 3 and think the manuscript will need to be reworked. We think that adjusting the conclusions will not impact the significance of the work, but is important, barring the generation of additional data to clarify the direction selectivity of these cells.

I have included, below the reviews, additional comments from the Academic Editor, which provide more specific advice that we hope will be helpful in guiding the revision. 

In light of the reviews and our Academic Editor's assessment, we are pleased to offer you the opportunity to address the remaining points in a revision that we anticipate should not take you very long. We will then assess your revised manuscript and your response to the reviewers' comments with our Academic Editor aiming to avoid further rounds of peer-review, although might need to consult with the reviewers, depending on the nature of the revisions.

**IMPORTANT: As you address these last points, we also ask that you address the following editorial requests: 

1) ETHICS STATEMENT: Please update the ethics statement in your manuscript to include the specific national or international regulations/guidelines to which your animal care and use protocol adhered. Please note that institutional or accreditation organization guidelines (such as AAALAC) do not meet this requirement.

2) BLURB: Please provide a blurb which (if accepted) will be included in our weekly and monthly Electronic Table of Contents, sent out to readers of PLOS Biology, and may be used to promote your article in social media. The blurb should be about 30-40 words long and is subject to editorial changes. It should, without exaggeration, entice people to read your manuscript. It should not be redundant with the title and should not contain acronyms or abbreviations.

3) METHODS: I see that you have included some of the methods section in teh supplement. Please move the methods to the main text. 

4) CODE: Per journal policy, if any code was generated to support the conclusions of your manuscript, we would require that you make it available without restrictions upon publication. Please ensure that any code is sufficiently well documented and reusable, and that your Data Statement in the Editorial Manager submission system accurately describes where your code can be found.

5) DATA: You may be aware of the PLOS Data Policy, which requires that all data be made available without restriction: http://journals.plos.org/plosbiology/s/data-availability. For more information, please also see this editorial: http://dx.doi.org/10.1371/journal.pbio.1001797

a. Supplementary files (e.g., excel). Please ensure that all data files are uploaded as 'Supporting Information' and are invariably referred to (in the manuscript, figure legends, and the Description field when uploading your files) using the following format verbatim: S1 Data, S2 Data, etc. Multiple panels of a single or even several figures can be included as multiple sheets in one excel file that is saved using exactly the following convention: S1_Data.xlsx (using an underscore).

b. Deposition in a publicly available repository. Please also provide the accession code or a reviewer link so that we may view your data before publication. 

>>Regardless of the method selected, please ensure that you provide the individual numerical values that underlie the summary data displayed in the following figure panels as they are essential for readers to assess your analysis and to reproduce it:

Fig 3F; 4C-F,H; Fig 5E,J,K-O;' Fig 6E-H; FIg S2 D-E; Fig S3B; Fjg S5; Fig S6D,F,H,J; Fig S7;

>>Please also ensure that figure legends in your manuscript include information on where the underlying data can be found, and ensure your supplemental data file/s has a legend.

>>Please ensure that your Data Statement in the submission system accurately describes where your data can be found.

**IMPORTANT - SUBMITTING YOUR REVISION**

*Resubmission Checklist*

*Published Peer Review*

*Blot and Gel Data Policy*

Sincerely,

Luke

Lucas Smith, Ph.D.

Senior Editor

PLOS Biology

lsmith@plos.org

REVIEWS:

Reviewer #2, Takeshi Yoshimatsu (note, Reviewer 2 has signed this review): In the revised manuscript, the authors addressed most of the concerns I initially had with the original version. The new manuscript has been much improved. I just have one more thing that I would like clarification on. The visual angle of the bar stimuli used in the two-photon calcium imaging is computed based on the visual distance of 0.36 cm and the bar size on the screen (line 1010 - 1011). However, it seems that, for two-photon imaging, larval fish were almost directly placed on the screen (line 1034 - 1035, 'Vertical distance from larvae lens to the screen was 0.5 mm'). If the calcium recordings were carried out in cells located in the dorsal region of the eye, shouldn't the visual distance be ~0.5 mm? This detail is particularly relevant because one of the paper's interesting findings is the difference in the bar size tuning between bhlhe22 and sox2 ACs. Adie from this, the experiments and data analysis are through, and the findings are very exciting.

Reviewer #3: The authors have responded in diligently to the previous reviews; however, I have one remaining concern pertinent to the basic conclusions. In my previous review I asked whether control data showing the magnitude of direction-selectivity indices (DSI's) recorded from non-directional cells under the same conditions was available. My interest was to see the variance and mean of DSI's in other non-DS retinal neurons, which could provide a basis for assessing the relative strength of DS responses in the two novel ACs. However, in this revised manuscript, responsive sox2+ and bhlhe22+ ACs were each divided in two groups; DS and non-DS cells. Surprisingly, the DS cells represent only a minor fraction of the total sample for each cell-type (Figs 5 & 6). 

These directional sox2+ and bhlhe22+ ACs are compared to the directional GABAergic/cholinergic starburst ACs reported in mammalian retinas. Starburst ACs uniformly show robust directional responses, so it is perhaps surprising that most of the novel sox2+ and bhlhe22+ ACs are non-directional. Thus, it would be more accurate to conclude from the data that the sox2+ and bhlhe22+ AC types in zebra fish are not direction selective, unlike their presumed mammalian counterparts. Consistent with this conclusion are the very modest effects that cell-specific KO of the ACs have on the OKR (Fig. 5).

The description of these novel ACs and their homology with mammalian AC types is interesting and significant. The study nicely documents functionally, morphologically, and genetically distinct GABAergic/cholinergic amacrine cells in a lower vertebrate retina. However, the evidence doesn't support the claim that they are functionally homologous to the mammalian cells or that they serve a major role in directional signaling. I would recommend that the authors revise the title, abstract, and discussion to reflect the findings more accurately.

---

COMMENTS FROM THE ACADEMIC EDITOR

Judging from the data provided by the authors, I am pretty sure their assertion of DS cannot be substantiated from these traces. In almost no case, the multiple repeats to the same stimulus give the same responses, and the "DS preference" only results from considering the average of some spontaneous events. Imagine a cell responds poorly to a given stimulus, but there is one or two spontaneous events that roughly coincide with when a stimulus was played. Then, on average, this cell would come out as directionally selective. However, these types of spontaneous events should have been filtered out. People typically compute some sort of quality metric for the repeatability of responses. I do not see such a thing used, and certainly, if used properly, it would kick out pretty much all of the trials shown in the pdf. Also, some of the "responses" look like motion artifacts (their kinetics are not plausibly explained by neural processing or calcium dynamics).

Therefore, based on the data I can see, I am with the reviewer: Either, the cells are not DS, or their DS-ness remains to be established with additional experiments and analysis.

In light of that I would strongly suggest the authors focus on the very nice description of these cells presence, organisation, genetics etc, and - importantly, the provision of the exceptionally useful transgenic lines that go with them, but keep the functional data very cautious, probably concluding that for now there is no strong evidence that there is DS, but importantly, that this doesnt mean that it is not there. It could also have been missed for technical reasons, and future work will be needed to carefully check this.

---

## [Editor Report · Decision Letter 3]

4 Jan 2024

Dear Dr He,

Happy New Year. Thank you for your patience while we considered your revised manuscript "Defining morphologically and genetically distinct GAGBergic/cholinergic amacrine cell subtypes in the vertebrate retina" for publication as a Research Article at PLOS Biology. This revised version of your manuscript has been evaluated by the PLOS Biology editors and the Academic Editor, who is now satisfied with the changes made in response to the last reviewer requests.

Based on our Academic Editor's assessment of your revision, we are likely to accept this manuscript for publication. However, before we can accept your study, we need you to address a number of data and other policy-related requests, detailed below. (These were included in our last decision letter, but I do not see the requested changes made, so I am including them here again).

**IMPORTANT: Please address the following editorial requests:

1) ETHICS STATEMENT: Please update the ethics statement in your manuscript to include the specific national or international regulations/guidelines to which your animal care and use protocol adhered. Please note that institutional or accreditation organization guidelines (such as AAALAC) do not meet this requirement.

2) BLURB: Please provide a blurb which (if accepted) will be included in our weekly and monthly Electronic Table of Contents, sent out to readers of PLOS Biology, and may be used to promote your article in social media. The blurb should be about 30-40 words long and is subject to editorial changes. It should, without exaggeration, entice people to read your manuscript. It should not be redundant with the title and should not contain acronyms or abbreviations.

3) METHODS: I see that you have included some of the methods section in the supplement. Please move the methods to the main text.

4) CODE: Per journal policy, if any code was generated to support the conclusions of your manuscript, we would require that you make it available without restrictions upon publication. Please ensure that any code is sufficiently well documented and reusable, and that your Data Statement in the Editorial Manager submission system accurately describes where your code can be found.

5) DATA: You may be aware of the PLOS Data Policy, which requires that all data be made available without restriction: http://journals.plos.org/plosbiology/s/data-availability. For more information, please also see this editorial: http://dx.doi.org/10.1371/journal.pbio.1001797

a. Supplementary files (e.g., excel). Please ensure that all data files are uploaded as 'Supporting Information' and are invariably referred to (in the manuscript, figure legends, and the Description field when uploading your files) using the following format verbatim: S1 Data, S2 Data, etc. Multiple panels of a single or even several figures can be included as multiple sheets in one excel file that is saved using exactly the following convention: S1_Data.xlsx (using an underscore).

b. Deposition in a publicly available repository. Please also provide the accession code or a reviewer link so that we may view your data before publication.

>>Regardless of the method selected, please ensure that you provide the individual numerical values that underlie the summary data displayed in the following figure panels as they are essential for readers to assess your analysis and to reproduce it:

Fig 3F; 4C-F,H; Fig 5E,J,K-O;' Fig 6E-H; FIg S2 D-E; Fig S3B; Fjg S5; Fig S6D,F,H,J; Fig S7;

>>Please also ensure that figure legends in your manuscript include information on where the underlying data can be found, and ensure your supplemental data file/s has a legend.

>>Please ensure that your Data Statement in the submission system accurately describes where your data can be found.

We expect to receive your revised manuscript within two weeks. 

*Published Peer Review History*

*Press*

Sincerely,

Luke

Lucas Smith, Ph.D.

Senior Editor,

lsmith@plos.org,

PLOS Biology

---

## [Editor Report · Decision Letter 4]

18 Jan 2024

Dear Dr He,

Thank you for the submission of your revised Research Article "Defining morphologically and genetically distinct GAGBergic/cholinergic amacrine cell subtypes in the vertebrate retina" for publication in PLOS Biology and thank you for addressing our last editorial requests in this revision. On behalf of my colleagues and the Academic Editor, Tom Baden, I am pleased to say that we can in principle accept your manuscript for publication, provided you address any remaining formatting and reporting issues. These will be detailed in an email you should receive within 2-3 business days from our colleagues in the journal operations team; no action is required from you until then. Please note that we will not be able to formally accept your manuscript and schedule it for publication until you have completed any requested changes.

**As you address the last formatting requests to come, we have one last editorial request that we ask that you attend to as well: 

1) Thank you for providing the data underlying your figures as a supplemental file (S3_Data). Can you please update the figure legends to reference this data file? For example, to each relevant figure legend, you can add the sentence "The data underlying this figure can be found in S3_data". 

PRESS

Sincerely, 

Luke

Lucas Smith, Ph.D., 

Senior Editor

PLOS Biology

lsmith@plos.org